# Sensitive neoantigen discovery by real-time mutanome-guided immunopeptidomics

Ilja E. Shapiro[1,2,3], Florian Huber [1,2,3], Justine Michaux[1,2,3] & Michal Bassani-Sternberg [1,2,3] ✉

Targeting cancer-specific HLA-peptide complexes is a promising approach in immunotherapy. Mutated neoantigens are excellent targets due to their immunogenicity and cancer-specificity. Mass spectrometry (MS)-based immunopeptidomics guides the selection of naturally presented immunogenic targets within the immunopeptidome, refining immunogenicity predictions. Implementation in clinical settings, however, must achieve global depth, capturing the entirety of the immunopeptidome, maintain high target sensitivity, and cater to scarce sample inputs and short turnaround time. Here, we present NeoDiscMS, an extension of NeoDisc that enables the acquisition of personalized immunopeptidomics data. Leveraging next-generation sequencing-guided real-time spectral acquisitions, NeoDiscMS maximizes sensitivity with minimal loss of global depth. Designed for effectiveness and ease of use, with minimal effort required for implementation, NeoDiscMS enhances the detection of peptides derived from tumor-associated antigens by up to 20% and improves confidence in neoantigen identification compared to the gold standard method. NeoDiscMS advances personalization in clinical antigen discovery with more confident neoantigen detection and easy implementation.

The accurate identification of immunogenic cancer-specific antigens presented by HLA complexes is instrumental to the development of cancer immunotherapy[1,2]. Most discovery approaches supporting neoantigen-based therapeutics rely solely on computational pipelines to predict immunogenic neoantigens from next-generation sequencing data of the patient's tumor and germline tissues[2–4]. Nevertheless, it has been repeatedly shown that only ~1% of somatic mutations in tumors induce spontaneous or vaccine-induced T cell responses[5,6]. Although the inclusion of machine learning classifiers improves the ranking of immunogenic neoantigens[7,8], the selection of candidate immunogenic neoantigens for inclusion in a personalized vaccine remains a challenge in the context of tumors with high mutational burdens[9]. Mass spectrometry (MS)-based detection of immunopeptides eluted directly from tumor tissues is the method of choice to simultaneously (1) deeply characterize the tumor antigenic landscape[10], (2) revealing novel targets like neoantigens[11], and (3)

characterize defects in antigen presentation[12], all of which are relevant clinical features[1,9]. MS-detected neoantigens have a substantially higher likelihood of being immunogenic than predicted neoantigens due to their confirmed presentation by tumor cells[11,13–18]. Besides the commonly known challenges that proteogenomic methods face[1], neoantigen detection by MS is further impeded by their low abundance[19,20]. In addition, the short turnaround time that is required for clinical vaccine manufacturing and the highly limited sample material pose important challenges when aiming to sensitively measure the global immunopeptidome while maintaining high sensitivity for selected neoantigens.

Typically, antigen discovery is conducted with TopN-based data-dependent acquisition (DDA). DDA is stochastic and aims to diversify what peptides are fragmented by utilizing dynamic exclusion, which prevents repeated fragmentation of the same precursor mass within subsequent acquisition cycles. While this approach increases

[1]Department of Oncology, University of Lausanne (UNIL) and Lausanne University Hospital (CHUV), Lausanne, Switzerland. [2]Ludwig Institute for Cancer Research, Lausanne Branch, Lausanne, Switzerland. [3]Agora Cancer Research Centre, Lausanne, Switzerland. ✉e-mail: michal.bassani@chuv.ch

coverage, it ultimately limits sensitivity for low-abundance targets. To improve sensitivity and address the limitation of co-isolated precursors (even in narrow-window MS acquisitions), previous studies applied chimeric spectrum deconvolution to identify tryptic peptides from low-input and single-cell proteomics samples[21–25]. However, so far, chimeric spectrum deconvolution has not been implemented in immunopeptidomics. Targeted immunopeptidomics strategies for neoantigen discovery exist, relying on heavy-labeled spike-in standards that enhance sensitivity[19,20]. However, synthesizing and conducting quality control on heavy-labeled peptides for each patient creates a significant bottleneck, making such strategies unsuitable for clinical workflows due to limited scalability and an inability to meet the short turnaround time required from biopsy to vaccine production. Peptide warehouse approaches can't alleviate these shortcomings, as neoantigens are unique to each patient. Additionally, a major concern is that synthesis products might include impurities of the unlabeled target peptide, potentially leading to sample contamination[20,26]. Spike-in free approaches that enhance sensitivity have been reported, but they (1) attempt to improve the sensitivity of TopN-based precursor selection with all its limitations, (2) focus on reproducing targets that were identified in a previous scouting run rather than discovering novel ones, (3) require advanced MS architectures, (4) demand manual intervention for data processing, and/or (5) don't optimize the trade-off between target sensitivity and global coverage[2,20,21,27–34].

Here, we present NeoDiscMS (an extension of NeoDisc[9]), a spike-in-free, scalable, and easy-to-implement targeted-DDA hybrid acquisition immunopeptidomic workflow that boasts enhanced sensitivity and accuracy for target peptide detection while minimizing the trade-off against loss of global immunopeptidome coverage. In NeoDiscMS, NGS-inferred in silico prioritized antigenic peptide candidates guide MS data acquisition by leveraging bespoke real-time peptide-to-spectrum matching filters (RTSf). RTSfs selectively trigger time-intensive, high-sensitivity scans for precursors with target-like features, a capability not previously applied to immunopeptidomics. In addition, chimeric spectrum deconvolution further enhances the depth of immunopeptidome analysis. We benchmark the performance of NeoDiscMS against DDA, either with or without an inclusion list, using an HLA-I immunopeptidome dilution series. To demonstrate the practical utility of NeoDiscMS we showcase the improved detection, in terms of reproducibility and identification confidence, of clinically relevant TAAs and mutated neoantigens in patient-derived primary melanoma cell lines and melanoma tissues. Importantly, data acquired with NeoDiscMS can be processed with commonly used search engines, facilitating seamless integration into existing workflows such as our clinical antigen discovery pipeline NeoDisc. We envision that NeoDiscMS may enhance the detection of immunopeptides associated with pathological conditions beyond cancer and can also be applied to the identification of other peptides outside the scope of immunopeptidomics.

## Results

### NeoDiscMS workflow layout
NeoDiscMS was implemented using a peptide database - in our case, an output of 1500 HLA-I-restricted predicted neoantigens or tumor-associated antigens generated by the end-to-end clinical antigen discovery pipeline NeoDisc[9], in the format of an inclusion list and a FASTA file containing each peptide as a separate entry (Fig. 1a). The MS acquisition of 3-s cycles was divided into three levels, in the following order of priority: the MS1 scan, the targeted branch scans, and the discovery (DDA) branch scans (Fig. 1b and Supplementary Fig. 1).

Each cycle starts with an MS1 scan. For a precursor in an MS1 scan to qualify for fragmentation with high-sensitivity scans in the targeted branch, it needs to match a precursor mass in the inclusion list. A matched precursor will first get fragmented in a scouting MS2 (sMS2) scan. The fragments of the sMS2 scan are matched in real-time, within

milliseconds, against predicted spectra in the database and are subsequently scored. For this, the cross-correlation[35] between the predicted fragments of peptides in the search space and the sMS2 spectrum is calculated. If the sMS2 scan yields a real-time search PSM that passes pre-defined quality thresholds, a time-intensive, high-sensitivity MS2 scan (hMS2) is conducted on that precursor (Fig. 1c). The sMS2 scan serves as a filter, minimizing the number of time-consuming subsequent targeted scans during a measurement. hMS2 scans can take up to five times longer than sMS2 scans and are defined by an increased AGC target, a higher maximum injection time, and stepped collision energies (Supplementary Table 1). Both sMS2 and hMS2 scans are intentionally not subjected to dynamic exclusion restrictions and MS1 minimum precursor intensity thresholds, enabling multiple MS/MS spectra to enhance identification confidence.

Following the targeted branch, the DDA branch is applied for the 3 s cycle's remaining time. To improve overall coverage, the discovery branch of our method uses wide isolation windows of 3.2Th for MS2 scans (dMS2) to increase the probability of co-isolating additional precursor masses. The obtained raw file is processed using chimeric spectra deconvolution with MSFragger's DDA+ mode[24], enabling peptide-to-spectrum matching (PSMing) with precursor masses that deviate from the isolated mass.

### Increased depth with wide isolation windows
To benchmark the performance of wide isolation windows and chimeric spectra deconvolution on immunopeptidomics data, we first tested different isolation window sizes with DDA, injecting the equivalent of 10 million JY cells per measurement, analyzed with MSFragger-DDA+. A total of 96% of the identified 8–14 mer peptides were predicted by MixMHCpred[36] as binders (%-rank <2) for each measurement, indicating that identification quality is consistent across conditions. Differences observed between isolation window widths pertained to the number of unique peptides identified. We selected 3.2Th because it most consistently yielded the highest number of identifications (Supplementary Fig. 2). This is in accordance with previous findings that optimal wide window size for increased depth with chimeric spectral deconvolution from low input samples remains narrow[24,25]. Next, to extend the benchmarking, we measured the HLA-I immunopeptidome of the B-cell lines JY and RA957[36], using either isolation windows of 1.2Th or 3.2Th, representing regular and wide isolation windows, respectively. This, for the equivalent 5 and 10 million cells per measurement. We then processed all data either with MSFragger-DDA or with MSFragger-DDA+ for peptide identification. For all measurements and with both MSFragger-DDA and MSFragger-DDA+, between 96–97% of the identified 8–14 mer peptides were predicted as binders (%-rank <2), the average peptide length lied between 9.4–9.6. Processing immunopeptidomic data with MSFragger-DDA+ instead of MSFragger-DDA resulted in a 2% to 36% increase in the number of unique peptides identified, with higher cell numbers contributing to a greater identification surplus (Fig. 2a). Notably, the RA957 samples exhibited larger increases compared to the JY samples. When comparing replicates of the same cell type and cell number, the change in identified peptides between isolation windows of 1.2Th and 3.2Th, processed with MSFragger-DDA+, ranged from 0% to 8%. The overlap coefficients for peptides identified by both MSFragger-DDA and MSFragger-DDA+ within the same raw file (top three gray dots in each panel of the lower row) ranged from 88% to 96% (Fig. 2a). Using MSFragger-DDA+, increasing the isolation window size from 1.2Th to 3.2Th resulted in a 7–9% increase in the fraction of scans yielding one or more PSMs (Fig. 2b). For MSFragger-DDA+ searched data, the fraction of peptides which were only identified thanks to chimeric spectrum deconvolution, meaning they never corresponded to the selected precursor mass to isolate, lies between 19–37%. Significantly, this fraction increases consistently by 10–11% when applying isolation

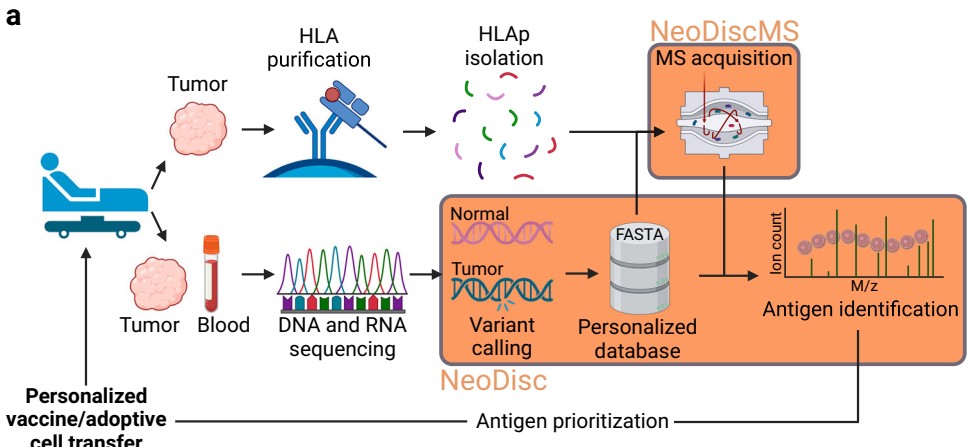

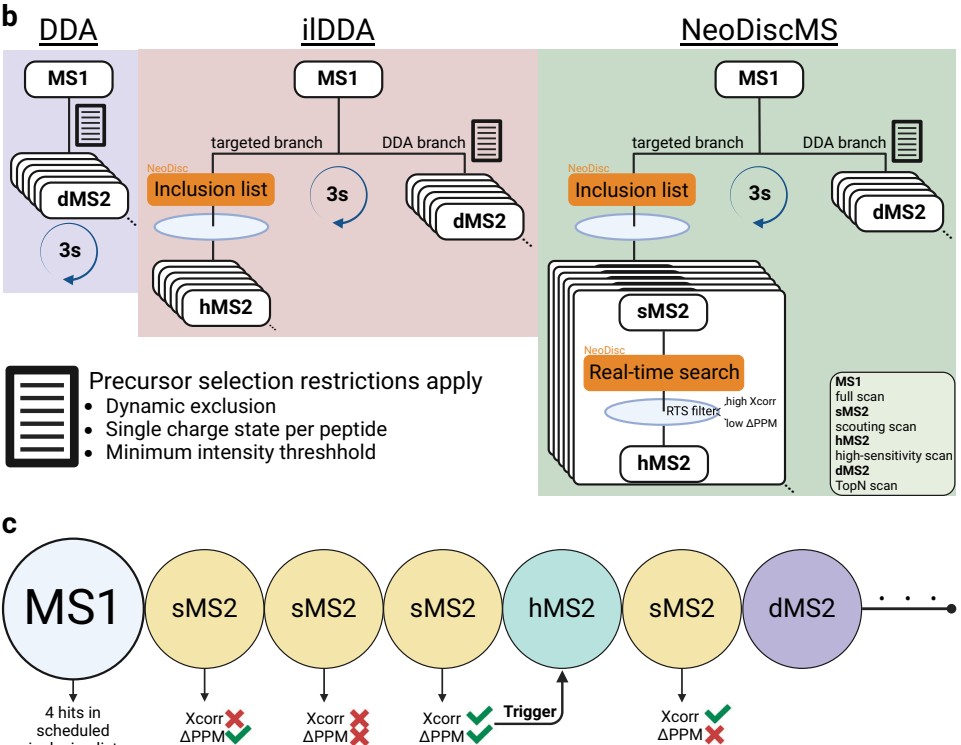

**Fig. 1 | Schematic overview of NeoDiscMS. a** Matched tumor and germline genome data, along with tumor transcriptome data, are processed by NeoDisc to create a tumor sample-specific personalized proteome reference, annotated with single-nucleotide polymorphisms, somatic mutations, which are then used for HLA binding and immunogenicity prediction of antigenic targets. From this prioritized list of peptides, NeoDisc generates a personalized inclusion list for NeoDiscMS acquisition of the matched tumor immunopeptidome sample. The MS data is then searched by NeoDisc against the personalized proteome to identify naturally presented antigenic peptides, refining the list to prioritize the most clinically relevant and likely immunogenic targets. **b** Comparison of DDA, DDA with an inclusion list (ilDDA) and NeoDiscMS methods. In NeoDiscMS, the MS acquisition of three

second cycles was divided into three levels, in the following order of priority: the MS1 scan, the targeted branch MS2 scans, and the discovery branch (DDA) MS2 scans. **c** The targeted branch of NeoDiscMS consists of scouting scans (sMS2) that get triggered by an MS1 precursor mass match and a retention-time-restricted scheduled inclusion list. sMS2 are searched against a database in real-time to assess if a high-sensitivity scan (hMS2) should be triggered. The metrics that constitute the real-time search are the cross-correlation between predicted and measured fragments (Xcorr) and the mass deviation of the predicted and measured precursor mass (ΔPPM). Schematics were created in Biorender ([https://BioRender.com/s85z530](https://BioRender.com/s85z530)).

windows of 3.2Th instead of 1.2Th for the same sample (Fig. 2c). To better understand the quality of PSMs gained with help of chimeric spectrum deconvolution, we inspected identified precursors that corresponded (isolated) and precursors that deviated from the isolated mass (co-isolated) of their respective MS2 scans as separate populations. Both populations had similar fractions of predicted binders to the respective HLA alleles and similar length distributions (Fig. 2d, e).

In a chimeric spectrum, the mass difference between the two peptides is defined by combinations of amino acids whose weights permit it to generate precursors that can be isolated through the same isolation window. Because the residuals of amino acid weights all lie in a narrow 0.1 Da range, we observed, as expected, that the absolute difference in mass (Da) between co-isolated peptides is often very close to multiples of 1 Da (when we filter for co-isolated identified peptides with the same charge which account for 92% of all cases; Fig. 2f).

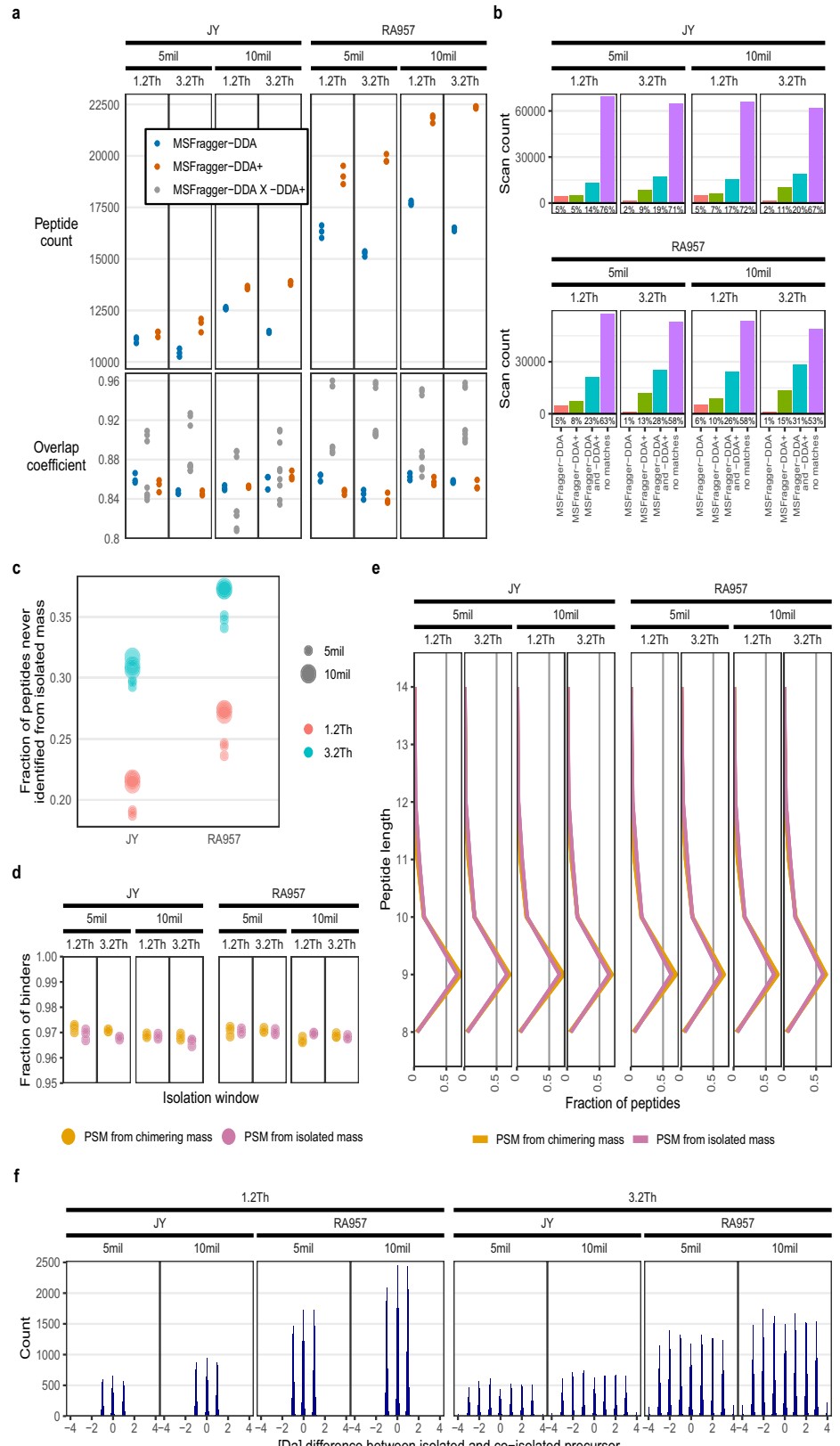

To conclude, the wide-window and especially the chimeric spectrum deconvolution approach provide increased global immunopeptidome depth over commonly used narrow-window scans in DDA while maintaining identification quality. Hence, the MS2 scans of the discovery branches in all the analyses below were conducted with an isolation window of 3.2Th, and the data were processed with MSFragger-DDA+. Importantly, the wide isolation window strategy impacts only the discovery branch, and users can implement NeoDiscMS without enabling it, should they choose to do so.

**Fig. 2 | Deconvolution of chimeric spectra. a** The upper panel depicts the number of identified unique stripped sequences in cell lines JY and RA957 immunopeptidome of injections of 5- or 10 million cells-equivalent, measured in DDA and with MS2 isolation windows of 1.2Th or 3.2Th ($n = 3$ per condition). The lower panel shows overlap coefficients between raw files and within or between processing modes. MSFragger can be operated with (MSFragger-DDA+) or without (MSFragger-DDA) taking chimeric spectra into account for finding PSMs. Wider isolation windows of 3.2Th and use of MSFragger-DDA+ increase the number of identified unique sequences, as demonstrated by searching the same raw files with either MSFragger-DDA or MSFragger-DDA+ (upper panel), resulting in an overlap coefficient of at least 88% (bottom panel; the top 3 gray dots for each sub-panel). **b** The

fraction of MS2 scans with at least one assigned PSM. Each raw file was processed with MSFragger-DDA or MSFragger-DDA+. For all raw files and processing modes, most scans with one PSM or more were matched by both processing modes. Percentages were averaged across technical replicates (TRs). **c** The fraction of immunopeptides that were identified by MSFragger-DDA+ exclusively as co-isolated PSMs. **d** The fraction of binders (%-rank <2; MixMHCpred) and **e** sequence length distribution among immunopeptides that were uniquely identified as chimeric PSMs or not **f** The co-isolated PSMs derive from precursors that deviate in mass from the isolated precursor with a clear pattern: mass deviation distribution reflects differences in multiples of 1 Da.

## Scalable enhanced sensitivity for targets

To illustrate the increased sensitivity of time-expensive scans for target discovery, as well as the capacity of RTSf to alleviate the burden on global immunopeptidome depth, we benchmarked the performance of DDA (the gold standard method for antigen discovery), DDA with an inclusion list branch (ilDDA) that triggers high-sensitivity scans, and NeoDiscMS (Supplementary Table 1 and Fig. 1b).

To accurately assess the performance of our target identification method, we compiled an inclusion list of ~1500 unique RA957-specific, HLA-A68:01-restricted, 8-11-meric peptide sequences (Fig. 3a). These peptides were consistently identified across RA957 samples and were predicted not to bind to any JY HLA-I alleles, showing a %-rank >10 for HLA-A*02:01, -B*07:02, and -C*07:02. (Fig. 3b). The list included methionine oxidation as a variable modification (Supplementary Fig. 3a) and all charge states from 1–3, yielding a list of 5046 precursors. To avoid having the targeted branch look for all targets throughout the entire gradient, our method relies on scheduled targeting. We predicted the retention time (RT) for each peptide to allow scheduling with a ±15 min window, which captures 75% of targets when comparing measured to predicted retention time, though at a remarkable inclusion list burden (number of scheduled precursors) across the retention time (Supplementary Fig. 3b, c).

We diluted the class I immunopeptide sample of RA957 cells in the class I immunopeptidome of JY cells in a dilution series that starts with RA957-to-JY dilution ratios of 1/16, 1/64, 1/256, and 1/1024, as well as a RA957-free JY immunopeptidome (negative control) (Fig. 3c). While the 1/1024 dilution simulates a setting with low abundance peptide targets that are hard to detect, the 1/16 dilution is most suitable to assess the scalability of NeoDiscMS when many detectable targets are present. We acquired data of all dilutions in DDA, ilDDA, and NeoDiscMS (Fig. 1b and Supplementary Table 1).

Overall, all measurements maintain between 96%–98% of binders, with an expected average peptide length of 9.4–9.5. The successful dilution is evident when inspecting the fraction of HLA-A*68:01 binders among all peptides (Supplementary Fig. 3d). None of the target peptides were detected in the JY-only negative controls across any of the acquisition modes. In all dilutions, NeoDiscMS identified between 0–7% more peptide targets on average than ilDDA, and between 16–51% more than DDA. Global depth of NeoDiscMS lied between 5–21% above ilDDA and 5–14% below DDA (Fig. 3d). With respect to the number of uniquely identified peptides we note that, on average, for every 1% of loss in global depth compared to DDA, we gain 2% in target sensitivity with ilDDA and 5.6% with NeoDiscMS. This difference in sensitivity trade-off quantifies the minimization of loss in global depth with help of RTSf very clearly.

In the dilution series, the 1/1024 dilution most closely simulates a real-world scenario of tumor-antigen discovery, with only a few detectable peptide targets. There, 18 peptide targets were identified consistently in all NeoDiscMS replicates, while this is only true for 9 peptide targets with DDA. This constitutes a 2-fold increase in the number of fully reproducible target peptide identifications. Conversely, 9 peptides were identified at least once with DDA and never with NeoDiscMS, while 19 peptides were detected at least once with

NeoDiscMS but not DDA, a 2.1-fold difference. In total, across three replicates each, 24% more peptide targets were identified with NeoDiscMS than with DDA (Fig. 3e). Detected peptide targets showed a higher average number of PSMs per target for NeoDiscMS than both DDA and ilDDA (Supplementary Fig. 3e). A closer examination of the 1/1024 dilution NeoDiscMS runs revealed that peptide targets which were not identified with DDA were distributed across all MS2 scan types and included low- to medium-intensity peptides. This shows that the enhanced target identification of NeoDiscMS derives from a combination of reduced ion sampling stochasticity and enhanced scan sensitivity, which sets targeted and DDA approaches apart (Supplementary Fig. 3f). For peptides in the 1/16 dilution, where the highest number of peptide target PSMs were detected, we observed a consistent increase in hyperscore and spectral angle when comparing hMS2 PSMs with dMS2 PSMs across the full intensity range (Fig. 3f).

Selective repeated fragmentation is one of the strengths of NeoDiscMS. This naturally raises the question of quantification reproducibility. To demonstrate the reproducibility of the MS1-based quantification, as provided by the FragPipe output, we focused on the 1/1024 dilution experiment measurements which represents a scenario of detecting low abundance targets. The variability between replicates is comparable between DDA and NeoDiscMS and between non-target peptide identifications (gray, $R^2 = 0.96$–0.97%) and target (red, $R^2 = 0.92$–0.99%; Fig. 3g). We observe slightly lower values in the technical replicates TR2 vs. TR3 NeoDiscMS comparison, though this is marginal considering the relatively lower number of target IDs. Repeated fragmentation via the targeted branch may open future opportunities to implement more accurate, advanced MS2-based quantification methods with NeoDiscMS data. However, a key caveat must be considered: hMS2 scans use different ion accumulation and collision energy settings compared to dMS2 and sMS2 scans (Supplementary Table 1). As a result, they introduce variations in both absolute fragment ion abundances and fragment intensity ratios, as exemplified by the target peptide EVILIDPFHK++ (Supplementary Fig. 4).

We concluded from our benchmarking with immunopeptidome dilutions that the targeted branch enables more sensitive and confident detection of peptides of interest compared to the discovery branch, and that NeoDiscMS does not compromise quantitative performance relative to DDA.

## Broader TAA and neoantigen discovery

To further characterize the utility of our method we acquired immunopeptidomics data with DDA and NeoDiscMS from three patient tumor-derived melanoma cell lines: Ml1, Ml2, and MI3 (Fig. 4a). As targets, we selected for each cell line NeoDisc-prioritized sample-specific mutated neoantigen peptides from pairs of tumor-healthy WES and tumor transcriptome data. In addition, to explore various potential sources of antigens we further extended the lists to well characterized TAAs, such as MAGEA3, MAGEA4, and MLANA, that were found to be expressed in the samples, and predicted TAA-derived peptides from in silico three frame translation products.

As expected, DDA provided the greatest immunopeptidome depth across all three cell lines, with NeoDiscMS showing a mean

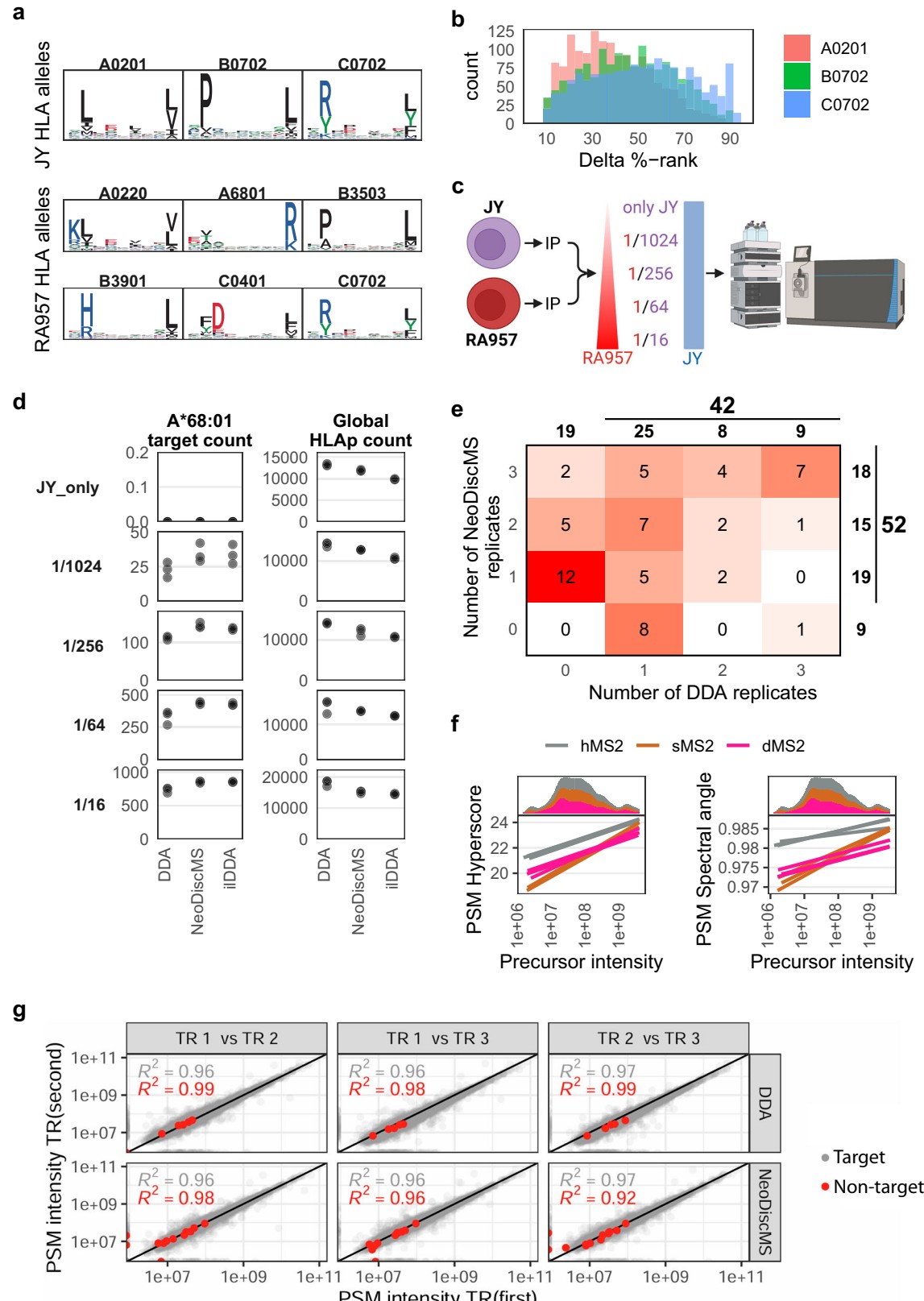

reduced depth compared to DDA by 21%, 13%, and 4% for MI1, MI2, and MI3, respectively. (Supplementary Fig. 5a). Overall, the immuno-peptidomes maintain between 94–95% of binders, and the average peptide length was between 9.3–9.6 (Supplementary Fig. 5b). We identified in total 49 TAA-peptides among them several confirmed immunogenic epitopes[37], and four mutated neoantigens out of which

three were previously demonstrated to be immunogenic in the patient[7,9] (Fig. 4a). Importantly, in each cell line, every target peptide identified with DDA was also identified with NeoDiscMS, apart from four targets, one of which detected three times and the others once across DDA replicates. Furthermore, the fraction of targets detected with NeoDiscMS that were identified in at least two out of three

**Fig. 3 | Benchmarking the global and target sensitivity of NeoDiscMS against DDA and ilDDA using an immunopeptidome dilution series. a** 9-mer motifs of immunopeptides that bind to the depicted HLA alleles. The key for diluting the class I immunopeptidome of RA957 in the class I immunopeptidome of JY is the distinctive A*68:01 binding restriction. We selected A*68:01-restricted peptides as targets because they can be clearly assigned to RA957 when the RA957 immuno-peptidome is diluted in the JY immunopeptidome. **b** Histogram of the binding affinity of A*68:01-specific targets to JY alleles as predicted by MixMHCpred. **c** Summary of the dilution series employed to evaluate the scalability and sensitivity of NeoDiscMS. **d** Count of identified A*68:01-binding target peptides and global depth for each step in the dilution series ($n = 3$ per condition). **e** Number of target peptides identified in either zero, one, two, or all three DDA (x-axis) or NeoDiscMS replicates (y-axis). Bold numbers show the sum of all targets in the respective row/column. Large bold numbers show the sum of all targets identified with either acquisition method. **f** Linear regression of all 1/16 dilution PSM hyperscores (left) and spectral angles (right), made with either dMS2, sMS2, or hMS2 scans. **g** Correlation of target and non-target precursor MS1 intensities between technical replicates (TRs) for 1/1024 dilution injections.

NeoDiscMS replicates but only in a single or none of the DDA replicates, was 16%, 24%, and 0% for Ml1, Ml2, and Ml3, respectively (Fig. 4b). On the protein level, NeoDiscMS can expand protein coverage substantially. For example, in Ml2, three out of five (THNVHEEKI, QHATIIHNL, THNIREEKI) SAGE1-derived peptides were identified consistently with NeoDiscMS but only once or never with DDA across all replicates (Fig. 4a).

## Confident neoantigen detection

To better characterize the improvement in peptide detection confidence, we closely investigated the underling qualities of RTS events that follow when the targeted branch is activated. When the precursor mass is detected in the scheduled RT window, a sMS2 scan is triggered. The sMS2 spectrum is then subjected to RTS. If the PSM passes the RTSf, we call it a RTS hit. A RTS hit consequently triggers the acquisition of a hMS2 scan. When a peptide precursor triggers an RTS event and that very peptide is also eventually PSMed with MSFragger in either sMS2 or hMS2, we call it a target identification (TID). The fraction of RTS events that resulted in an RTS hit was 12%, 8%, and 8% on average for Ml1, Ml2, and Ml3, respectively. More than 85% of the inclusion list hits during a measurement do not pass RTSf, indicating that they are isobaric non-targets (Supplementary Fig. 5c). The fraction of RTS hit events that resulted in a TID was 1%, 6%, and 2% on average for Ml1, Ml2, and Ml3, respectively (Supplementary Fig. 5d) and all TIDs occurred in RTS hit events (Supplementary Fig. 6). Across samples, at least 90% of all RTS hit events that yielded a TID fit into a ΔPPM window smaller than 1 (Supplementary Fig. 5e). Among all RTS events with TIDs, the majority were PSMed in both sMS2 and hMS2 scans. Ml2 particularly benefits from hMS2 scans, as the number of uniquely hMS2-derived TIDs is 60% higher than uniquely sMS2-derived TIDs (Fig. 5b). The RTS result of TIDs that were uniquely identified with hMS2 scans remained within the Xcorr range [0.4, 2.7] (Supplementary Fig. 6). TIDs that succeeded for sMS2 scans and hMS2 scans within the same RTS events are the best possible measure for the immediate gain in sensitivity between these two scan types. Particularly for low scoring sMS2 TIDs, hyperscores and spectral angles of the same precursor PSM increased for subsequent hMS2 spectra (Supplementary Fig. 7).

The MCTS1[D130N]-derived neoantigen YPAAVNTIVAI found in Ml2 showcases the clear advantages of implementing NeoDiscMS for the confident discovery of clinically relevant mutated neoantigens. Based on predictions, NeoDisc ranked this neoantigen at position 186 among predicted HLA-I neoantigens, while an autologous TIL-based immunogenicity assay confirmed its immunogenicity[7,9]. In a hypothetical patient case where neoantigen targets for a personalized cancer vaccine are identified solely through mutanome predictions, a neoantigen ranked as low as 186 would generally be excluded from selection, as typically up to 40 top ranked neoantigens are targeted[38–40]. However, in a clinical discovery effort combining immunopeptidomics with mutanome predictions, a robust MS support for this immunogenic neoantigen would justify its inclusion. Conversely, without solid MS backing, including such a low-ranked peptide over a higher-ranked neoantigen would not be justified.

YPAAVNTIVAI was identified in all immunopeptidomic replicates with DDA, however, with only up to two PSMs per replicate.

Conversely, with NeoDiscMS, repeated RTS hit events across successive acquisition cycles resulted in 9 to 13 PSMs per replicate, of which 5 to 7 of YPAAVNTIVA PSMs derived from hMS2 scans (Fig. 5a–c). Despite having a spectral angle of 0.9299, the best dMS2 YPAAVNTIVA PSM in the DDA contained only 7 fragments (b5–b9, y1 and y2), where none of the first four peptide bonds is covered by a fragmentation site (Fig. 5d). Across NeoDiscMS runs, the best dMS2, sMS2, and hMS2 had spectral angles of 0.9635, 0.9766, and 0.995. The best hMS2 PSM matched 12 fragments, (b2–b10, y1–y3), resulting in accurate and confident peptide identification. To conclude, Neo-DiscMS resulted in consistent and confident YPAAVNTIVAI neoantigen detection with high spectral angles and number of matched fragments.

## Direct clinical application

To demonstrate its clinical applicability, we utilized our clinical immunopeptidome pipeline[9] to extract the immunopeptidome from three distinct tumor lesions (Ti1, Ti2, Ti3) of a uveal melanoma patient, with each NeoDiscMS analysis performed on the equivalent of 10 mg of tissue. We selected the top 500 predicted neoantigens as well as the top 518 predicted TAA-derived peptides as targets in a single list that was used for NeoDiscMS measurements of all three tissue samples. Our clinical routine immunopeptidomics acquisition scheme includes both DDA and DIA measurements allowing sensitive peptide-centric DIA searches against a sample-specific hybrid spectra library generated from spectrum-centric analysis of the DIA file combined with the library of generated from the DDA file[9,41]. Therefore, we acquired one technical DIA and NeoDiscMS acquisition for each tissue sample, using a personalized NGS-guided target database. With NeoDiscMS for each tissue sample individually, we identified 14,797, 16,033 and 11,968 unique peptides at an average length of 9.4 and with 95% of binders for Ti1, Ti2, Ti3, respectively (Fig. 6a). For all three tissues, the number of identified peptides with spectrum-centric searches of DIA data (DIA-Umpire) amounts to less than 50% of the depth that was achieved with a corresponding NeoDiscMS search. Then, we used spectral hybrid-libraries (generated from NeoDiscMS and DIA data), to search the DIA files and identified and quantified 12,917, 14,165 and 10,726 unique peptides in Ti1, Ti2, Ti3, respectively (Fig. 6a). An examination of the shared and sample-specific peptide sets revealed differences in peptide presentation across the three samples. The fraction of peptides common to all three lesions is 31% in the NeoDiscMS search and 36% in the hybrid library DIA search comparison (Fig. 6b). As the tumor mutation burden was expectedly low in this uveal melanoma tumor lesions (2.601, 2.82, and 3.615 somatic mutations/Mb for Ti1, Ti2, and Ti3, respectively), no neoantigens were identified in any of the samples. Yet, nine TAA-derived peptides were identified in total, derived from MLANA, TYR, and SAGE1, with peptide MPREDAHF from MLANA detected in the DIA files of Ti1, Ti2, and Ti3, and peptide YRALMDKSL from MLANA in Ti1, only when analyzed against the hybrid library and not with the spectrum-centric approach. Hence, we demonstrated that NeoDiscMS can be seamlessly combined with DIA to create sample-specific hybrid libraries enriched with high-confidence MS2 spectra, for the sensitive detection and quantification of the global immuno-peptidome as well as target peptides (Fig. 6a, c).

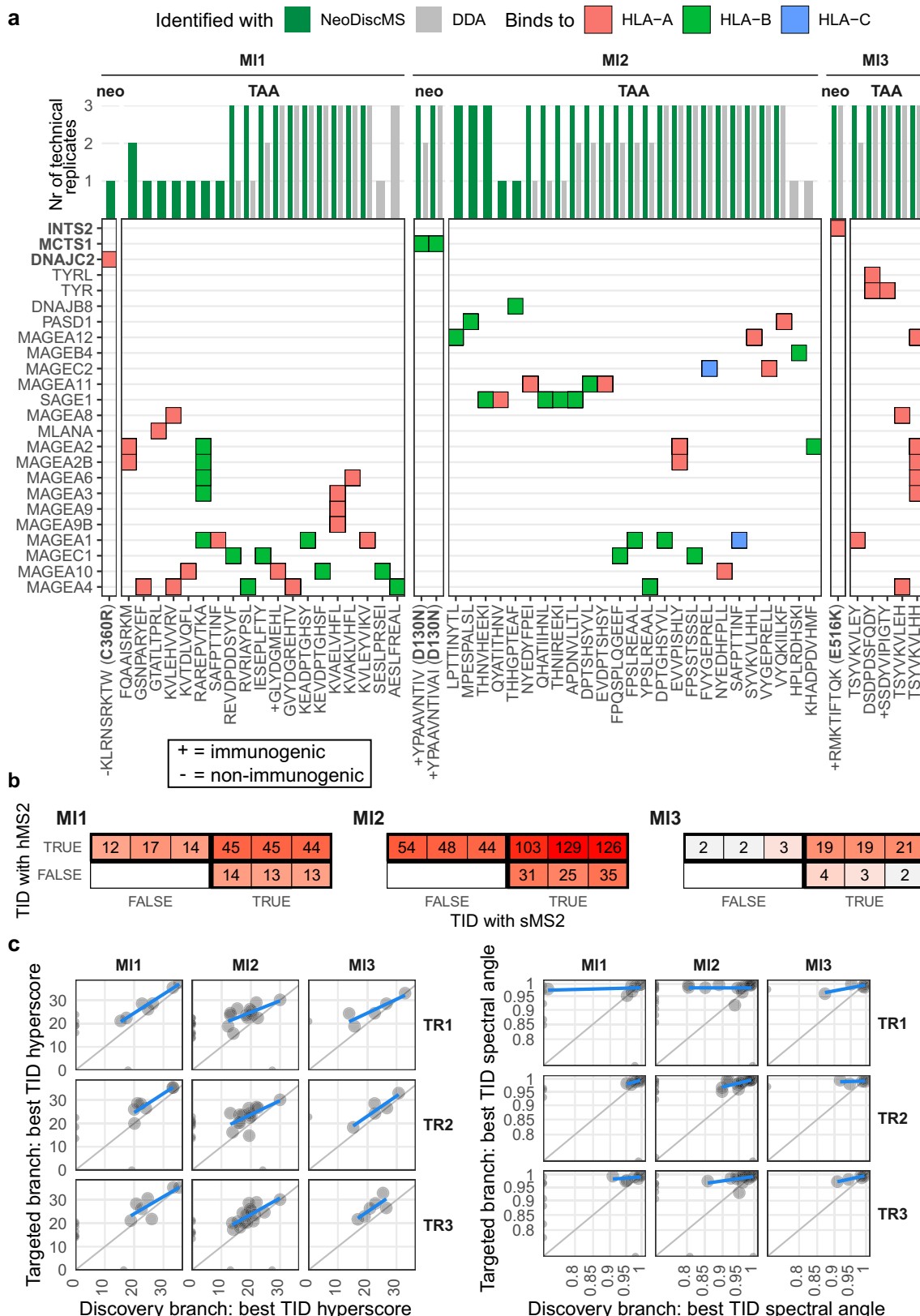

**Fig. 4 | Accurate and sensitive detection of neoantigens and TAA-derived immunopeptides with NeoDiscMS compared to DDA in three melanoma derived cell-line. a** The individual identification count (upper panel) of target peptides (*x*-axis) among DDA and NeoDiscMS replicates (*n* = 3 per condition). Gene mappings (*y*-axis) order the identified neoantigens at the top. **b** The number of target identifications (TIDs; identification of precursor that triggered the targeted branch and was identified in either sMS2 and hMS2) made with either sMS2, hMS2 or both within an RTS event. Three adjacent blocks separated by thin black lines represent replicates 1,2, and 3. **c** The best TID hyperscore (left) or spectral angle (right) of each identified target peptide compared to the same peptide's best hyperscore or spectral angle in the discovery branch.

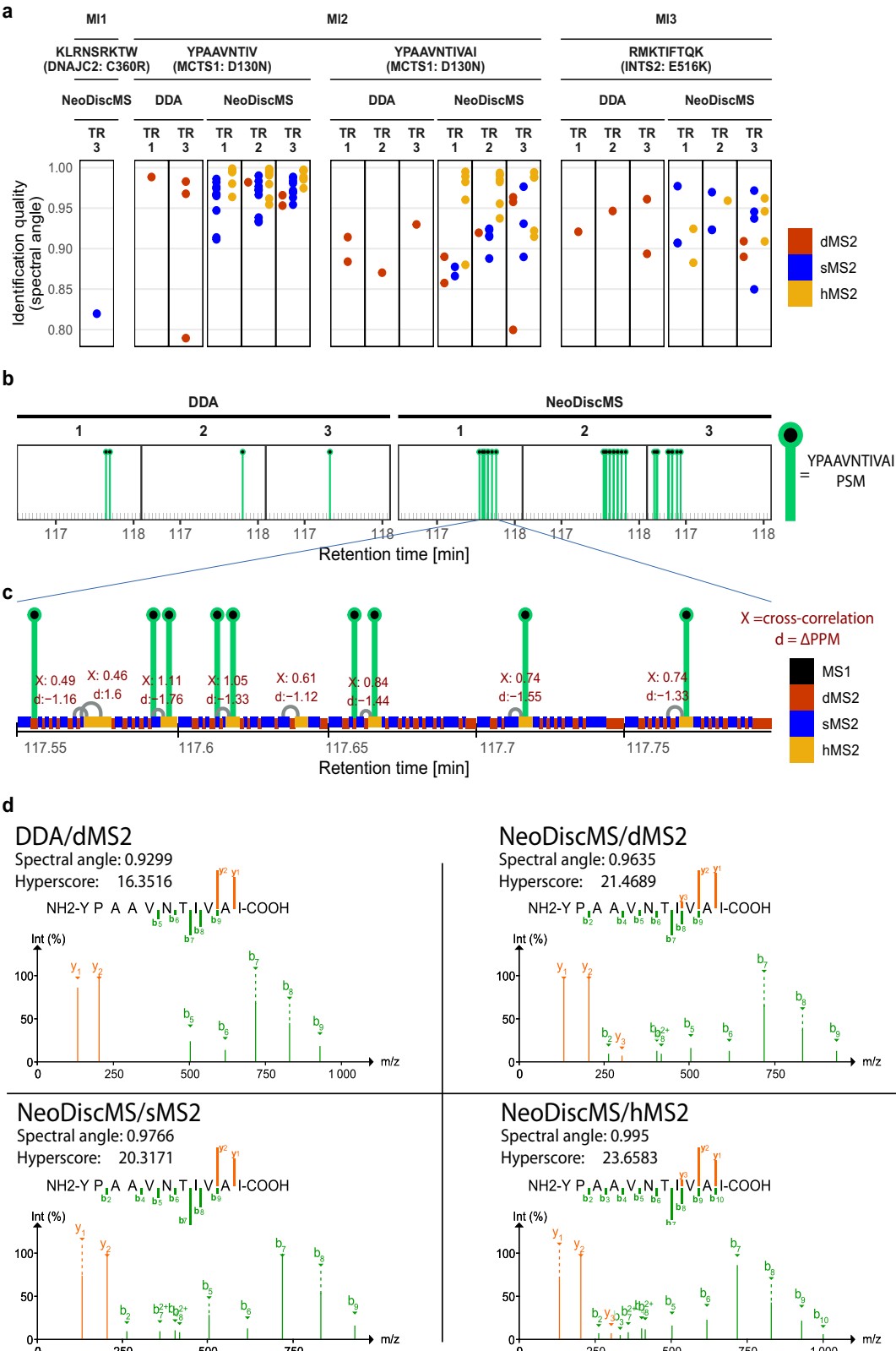

**Fig. 5 | NeoDiscMS improves the confidence of neoantigen identifications.**
**a** Each dot represents a PSM for the corresponding neoantigen (TR = replicate).
**b** The retention time and identification rate of example neoantigen YPAAVNTIVAI in all three DDA and NeoDiscMS replicates of MI2. **c** Zoom-in of the retention time window in NeoDiscMS technical replicate 1 of (**b**). Real-time search events where the real-time search filters are passed (RTS hit events) connect the respective sMS2

and hMS2 with a linker (gray arches). The real-time search parameters of each RTS hit event hovers above the respective linker. **d** The top YPAAVNTIVAI PSMs, based on spectral angle, for each scan type in DDA and NeoDiscMS demonstrate that hMS2 scans offer important improvements, including an increased number of identified fragments.

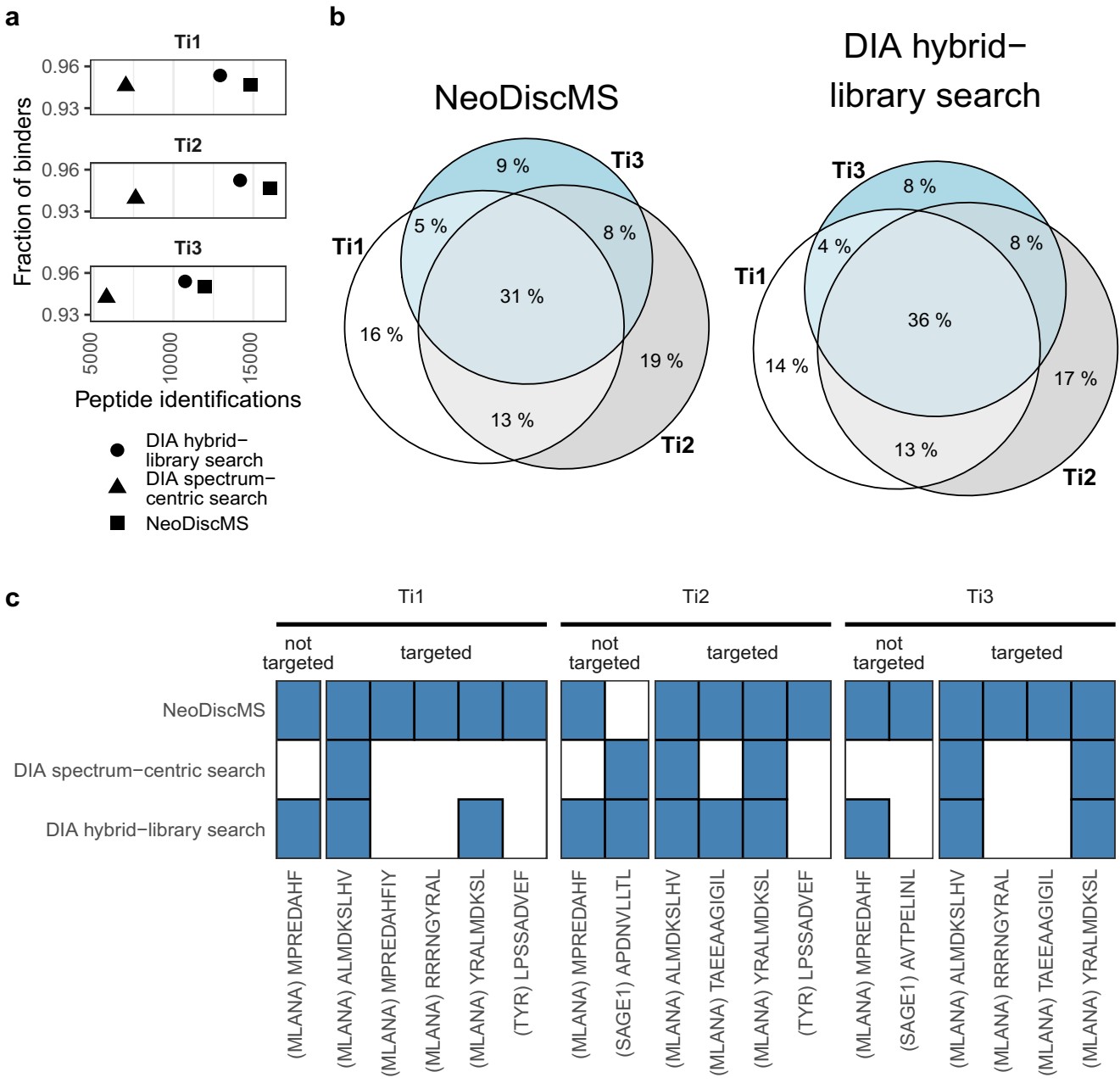

**Fig. 6 | NeoDiscMS can be integrated into DIA library search strategies for tumor biopsies. a** The analysis of three low-input uveal melanoma tissues (10 mg/injection) of the same patient (Ti1, Ti2, and Ti3, respectively), reveals a global depth of 12,316, 13,490, and 10,232 immunopeptides (binders only) with a DIA-NeoDiscMS hybrid-library search, with more than 95% of the identified peptides predicted as binders by MixMHCpred (%-rank <2). **b** Venn diagram describing the intersection of presented immunopeptides between Ti1, Ti2, and Ti3 with either a spectrum centric NeoDiscMS search (left) or a DIA-NeoDiscMS hybrid-library search (right). **c** Identified TAAs with either NeoDiscMS or spectrum-centric searches of DIA data, or a peptide-centric search of DIA with a DIA-NeoDiscMS hybrid-library.

## Discussion

In this study we present a new immunopeptidomic MS acquisition method that has been developed specifically to overcome critical challenges related to the discovery of clinically relevant antigens. Tumor-specific antigens are often present at low abundance, yet in clinical settings, they require rapid, robust, and accurate detection to enable their use as immunotherapeutic targets. Recently we reported NeoDisc, a clinical end-to-end computational pipeline that combines genomics, transcriptomics, and MS-based immunopeptidomics for the prediction and direct identification of immunogenic tumor-specific HLA-I and -II antigens from multiple sources. So far, once the NGS and MS-immunopeptidomic files have been generated, they were subjected to the NeoDisc computational integration. Here, we propose an alternative approach. Matched tumor and germline genome data, along with tumor transcriptome data, are processed with NeoDisc to create a tumor sample-specific personalized proteome reference. The reference is annotated with single-nucleotide polymorphisms and somatic mutations, which are then used for HLA binding and machine learning-based immunogenicity prediction of antigenic target peptides. From this prioritized list of targets, NeoDisc generates a personalized inclusion list for NeoDiscMS acquisition of the MS analysis of the matched tumor immunopeptidome sample, followed by a DIA acquisition. The immunopeptidomic MS data is then searched by NeoDisc against the personalized proteome to identify naturally presented peptides, refining the list to prioritize the most clinically relevant and likely immunogenic targets. This streamlined process

maintains a 2-week turnaround from biopsy reception to the compilation of a ranked list of targets, for example for neoantigen manufacturing.

NeoDiscMS is a scalable, spike-in-free immunopeptidomic discovery workflow that can also operate independently of NeoDisc, using any list of target peptides of interest. Importantly, the key use case for NeoDiscMS are highly limited biopsies with strict time limits for data acquisition, whereby both target sensitivity and global depth are required. Samples are prepared following standard immunopeptidomic protocols, and, unlike other targeted approaches, no prior MS measurements of the investigated samples are required. This allows the entire sample to be utilized at once, enhancing detection and decreasing turnaround time.

During acquisition, the separation of discovery and targeted branch as well as fully customizable parameters for each scan type, facilitates maximal flexibility to optimize the trade-off between target sensitivity and global depth. The edge of NeoDiscMS over ilDDA lies in more efficient application of high-sensitivity scans. Long injection times on every analyte that is isobaric to a target within the scheduled RT window becomes increasingly troublesome as inclusion lists are scaled: the number of targets as well as the scheduled RT window size are critical for inclusion list burden. NeoDiscMS significantly reduces time spent sampling off-target ions. This leaves more time within an acquisition cycle to sample ions of other target masses or with the discovery branch. An additional advantage of NeoDiscMS is its ability to sample the same precursor up to three times per cycle, compared to a maximum of two times in ilDDA. Each MS2 scan contributes a spectrum that can yield a high-quality PSM. This increased sampling frequency explains the higher global depth and the equal or slightly improved target sensitivity observed with NeoDiscMS relative to ilDDA.

The decision tree of the targeted branch we present is simple (sMS2−RTSf−hMS2) and in the future, sophistication of RTS event designs and sMS2-hMS2 dependencies are expected to further boost the sensitivity while minimizing loss of global depth. Selective high-sensitivity scans can expand beyond the optimized ion accumulation and collisional energy parameters that we showcase to, for instance, ion mobility parameters[20,42]. Also, narrower isolation windows could provide cleaner spectra with less interfering peaks, at the cost of sensitivity or injection time. Dynamic definition of data aquisition parameters with help of sMS2 scans, just like the refinement of real-time search evaluation parameters, bear great potential to further advance target identification. Acquiring multiple scans of the same precursor of interest within and between cycles offers novel opportunities for peptide quantification and generation of consensus spectra. Additionally, precursor selection parameters like adjusted minimal intensity thresholds or better integration of inclusion list and RTS FASTA as well as optimized target precursor selection offer further possibilities for refinement.

The targeted branch of NeoDiscMS is committed to the holy grail of clinical immunopeptidomics: the direct identification of cancer-specific antigens. Yet, we would like to highlight the substantial importance of the discovery branch that provides global depth in a DDA-like fashion. Firstly, the global immunopeptidome informs us about dynamics of antigen processing and presentation hotspot features, as well as defects in antigen processing and presentation, which can directly inform immunogenicity predictions and neoantigen prioritization[7,9,10,43–45]. Naturally, the discovery branch can also contribute additional spectra of the targets as well as of other non-targeted exploratory antigen candidates. One great advantage of mass spectrometry data is that it can be re-processed with new search settings and software to extract new information or build large databases[36,46,47]. The discovery branch provides the bulk, in absolute terms, of high-quality spectra in a NeoDiscMS data file, being "TopN"-sampled, and provide utility for this purpose. Furthermore, the PSMs

that the discovery branch yields during data processing enables statistically solid FDR calculations[48] by providing highly diverse target and decoy matches, without being restricted to only sampling ions that correspond to the ~5000 masses in the inclusion list of the targeted branch. Additionally, high-quality PSMs by the discovery branch are very likely to be identified and instrumental for effective feature extraction in rescoring algorithms such as MSBooster[47]. As such, the discovery branch provides not only additional depth but also enables the effective bioinformatic data processing with commonly used search engines so that targets can be identified from the targeted branch with higher sensitivity.

Being able to process data with popular proteomics tools such as MSFragger[49], is crucial for the user-friendliness of the NeoDiscMS approach. The application of chimeric spectrum deconvolution and wide isolation windows in the discovery branch boosts global coverage while maintaining high identification quality. As such, we show that chimeric spectrum deconvolution with MSFragger-DDA+ can be applied to immunopeptidomics data to great benefit. While NeoDiscMS can be seamlessly processed by MSFragger, we also show how NeoDiscMS offers opportunities for more extensive data processing for MS2-based quantification of precursors that were identified as the isolated mass multiple times.

Our data shows that the targeted and the discovery branches contribute improvements to NeoDiscMS compared to conventional DDA approaches. The increased isolation window size in combination with chimeric spectral deconvolution results in increased immunopeptidome depth, as shown in a direct comparison of regular and wide window isolation windows on the same sample, analyzed with MSFragger-DDA or MSFragger-DDA+. With the help of JY, RA957, and melanoma-derived cell lines, we show that the targeted branch improved target identification significantly, consistently, and in a scalable manner. By examining RTS event dynamics, we show how and at what rate sMS2 and RTSf selectively trigger high-sensitivity scans. This analysis reveals the individual contributions of sMS2 and hMS2 scans. We highlight YPAAVNTIVAI, an immunogenic neoantigen that is a direct beneficiary of the increased sensitivity that NeoDiscMS offers: YPAAVNTIVAI's identification confidence clearly and consistently improves with NeoDiscMS compared to DDA.

We demonstrate the clinical utility of NeoDiscMS with a set of three low-input tumor biopsies. By simulating a clinical setting, we achieve substantial depth as well as the identification of TAA-derived peptides and demonstrate interoperability with DIA library searching strategies. Spectrum-centric searches of DIA data yielded substantially fewer peptide identifications compared to NeoDiscMS (Fig. 6a). This makes NeoDiscMS a suitable candidate for library building to fully harness the power of DIA library searches. Alternatively, experimental libraries can be replaced with computationally predicted libraries. This approach holds great promise, yet it's performance to provide more depth in immunopeptidomics compared to experimental libraries is, to date, situational[41,50,51]. Also, accurate FDR calculations for large-scale predicted libraries for peptide-centric searches might face processing-specific challenges that need to be evaluated and addressed, just as it is the case for DDA database searches[52,53]. Besides the construction of experimental libraries, another key complementary feature of NeoDiscMS to DIA is that it provides low-complexity spectra with its targeted branch, compared to DIA MS2 spectra. This is crucial when identifying peptides for clinical prioritization, where manual inspection of signal and background fragments of candidate PSMs is common. The manual inspection and validation of a PSM from DIA MS2 spectra is virtually impossible due to its massive complexity, even for the most advanced experts. If data acquisition approaches that are still being developed, such as narrow-window DIA[54], will shift this balance remains to be seen.

At the time of writing, the instruments equipped with vendor-supported acquisition method editors that enable implementation of

NeoDiscMS are the Orbitrap Eclipse and Orbitrap Ascend (part of Thermo Fisher Scientific's Tribrid Mass Spectrometer series). We envision broader vendor adoption of this approach in the future, as well as the integration of advanced RTS capabilities that extend beyond immunopeptidomics applications.

NeoDiscMS is, to our knowledge, the first personalized immunopeptidomics data acquisition strategy tailored to the needs of a proteogenomics-based clinical T-cell antigen discovery pipeline. Besides analytical aspects, its guiding design principles were oriented towards modularity, flexibility, and user-friendliness. We conclude that NeoDiscMS advances proteogenomic-based antigen discovery with all its clinical benefits for the development of cancer therapies. Generation of NeoDiscMS files (target selection, FASTA database, and inclusion list) is integrated within NeoDisc v1.7.1 and is available through https://neodisc.unil.ch/.

## Methods

### Patient samples, cell lines, and cell culture
An informed written consent was given by the participants, according to the requirements of the local Ethics Committee institutional review board (Ethics Commission, Commission cantonale d'éthique de la recherche sur l'être humain (CER-VD) Centre hospitalier universitaire Vaudois, CHUV). Ti1, Ti2, and Ti3 samples used in this investigation were collected from the patient during the screening process for inclusion in a Phase 1 trial (NCT04643574), following enrollment in a research protocol approved by the local Ethics Committee (Commission cantonale d'éthique de la recherche sur l'être humain (CER-VD); BASEC ID 2017-00305). Tumor cell lines were generated as described in Huber et al.[9]. Ml1 and Ml2, including immunogenicity assessments and NGS data, are the same as described in Müller et al.[7], where they equate to patient1 and patient3, respectively. Ml3, including immunogenicity assessments, corresponds to Mel-4 P1 HLA-I high cells, stimulated with 100 U/mL of Interferon-γ (Miltenyi Biotec, ref. 130-096-482), as described in Huber et al.[9]. HLA-restricted TAA immunogenicity assessments are derived from CEDAR[37].

All cells were expanded in cell culture medium (Gibco, ref. 61870-010) with 10% of heat-inactivated FBS (Gibco, ref. 10437-028) and 100 U/mL P/S (BioConcept, cat No. 4-01F00-H). To harvest the JY (ATCC, ref. 77441) and RA957 cell lines we collected the cell culture medium containing the cells in suspension, removed the medium, and washed cell twice in PBS (Bichsel, ref. 100 0 324). To harvest melanoma cell lines we removed the cell culture medium, washed the cells with PBS, detached cells from the flask surface with Trypsin (BioConcept, ref. 5-51F00-H), inactivated trypsin with cell culture medium, and washed the cells twice with PBS. We stored cells in pellets of $10^8$ at −80 °C.

### Immunopeptide enrichment
For immunopeptide enrichment we harvested W6/32 antibodies from HB-95 hybridoma cells (HB-95, ATCC), cross-linked them with DTT to Sepharose beads (Invitrogen, ref. 101042). We then lysed our samples with 0.25% sodium deoxycholate (Sigma-Aldrich, ref. 30970), 0.2 mM IAA (Sigma, ref. I6125-5g), 1 mM EDTA (Thermo Fisher Scientific, ref. 15575-038), 1:200 PIC (Roche, ref. 04693132001), 1 mM phenylmethylsulfonyl fluoride (Roche, ref. 10837091001), 1% octyl-beta-D glucopyranoside (Sigma-Aldrich, ref. 08001) in PBS for 1 h at 4 °C. Cell lines were just resuspended in lysis buffer, tissue samples were first homogenized with a bead homogenizer (Tissue Lyser II, Qiagen) for 1 min at 30 Hz. We then centrifuged lysates for 50 min at 20,000 RCF at 4 °C. To isolate HLA-peptide complexes, we applied the workflow presented in Chong et al.[55]. Shortly, 96-well plates were conditioned and mounted with 75 µl of cross-linked beads. We then added each sample's lysate to a well. After the lysate drained, we washed each well with varying salt concentrations and eluted the captured HLA-peptide complexes with 1% TFA (Merck Millipore, ref. 1082620100). Then we

desalted and isolated immunopeptides with the help of C18 columns, using 28% ACN (Biosolve, UN 1648), 0.1% TFA as an elution buffer. The peptides eluted from the C18 column were dried in a vacuum concentrator (Concentrator plus, Eppendorf) and stored at −80 °C. Prior to MS analysis, peptides were re-suspended in 30 µl (cell lines) or 18 µl (tissue samples) 0.1 %TFA. MS iRT kit peptides (Biognosys, ref. Ki-3002-2) were spiked into the peptidomic samples according to the supplier's instructions.

For all experiments with cell lines, we isolated immunopeptides from samples of $10^8$ cells each. For all experiments with tissues, we processed ~60 mg of each tissue. For cell lines, we injected an equivalent of $5*10^6$ or $10^7$ cells (Supplementary Data 1). For tissue cells, we injected equivalents of ~10 mg of tissue. For the dilution experiment, RA957 immunopeptidome was diluted in the JY immunopeptidome at the ratios of 1/1024, 1/256, 1/64, and 1/16. In addition, we included a negative control sample with only the JY peptides. Importantly, all samples contained equal amount of JY peptides.

### WES and RNAseq library preparation and sequencing
WES and RNAseq libraries were prepared as described in Huber et al.[9]. Briefly, DNA was extracted with the commercially available DNeasy Blood and Tissue Kit (Qiagen, ref. 69504) according to the manufacturer's protocols. RNA was extracted by phase separation using Trizol (Thermo Fisher Scientific, ref. 15596-026) and chloroform (Sigma-Aldrich, ref. C2432-25ML) and centrifugation at 12,000 × $g$ for 15 min. Then, the aqueous phase containing the RNA was collected, mixed with 70% ethanol and loaded on an RNeasy mini spin column from the total RNA isolation RNeasy mini kit (Qiagen, ref. 74104). The rest of the protocol was performed according to the manufacturer's instructions (including DNase I (Qiagen, ref. 79254) on-column digestion). WES and RNAseq libraries were prepared and sequenced at Microsynth using the Agilent SureSelect XT Human All exome V7 kit (Agilent, ref. 5191-4028) and the Illumina Truseq stranded mRNA reagents (Illumina, ref. 20020594). The sequencing was performed on the NextSeq 500/550 system.

### NeoDisc pipeline parameters
NeoDisc v1.7.0 was used for neoantigen discovery and prioritization[9]. The pipeline was run in fastq mode using default parameters for single-sample analysis on paired germline and tumor whole-exome sequencing data, along with matched tumor RNA sequencing data. Peptide lengths for HLA-I and HLA-II predictions were defined in the configuration file as 8–12 and 12–15 amino acids, respectively, with binding affinity thresholds set to ≤2.0% for both HLA-I and HLA-II predictions.

Following single-sample analyses, results were combined using NeoDisc's MergeSamples tool to facilitate multi-sample integration and prioritize tumor-specific antigens.

### NeoDiscMS workflow
The NeoDiscMS approach require the generation of a list of target peptides. In this study we tested the performance with ~1500 targets sequences. In general, any peptide sequence can be included as target.

For the JY-RA957 dilution experiment, we selected RA957-specific HLA-I bound peptides identified in previously published eight raw files of RA957 immunopeptidome samples[55]. We processed the raw files with MSFragger-DDA+ and filtered all identified peptides to retain peptides of length 8–11, that occurred in all eight measurements, were predicted by MixMHCpred[36] (v2.3) to be A*68:01 binders with binding %-rank <2, and did not have a %-rank <10% for any other RA957 or JY alleles. This yielded 1525 peptide targets.

For each of the melanoma cell lines Ml1, Ml2, and Ml3, 500 predicted neoantigens with %-rank <3% were selected based on their prioritization by NeoDisc[9]. In addition, we supplemented the inclusion lists with peptides derived from three-frame translation of TAA genes (Supplementary Data 2). Here, we filtered all 8–14mers so that each had

%-rank of <2 for at least two HLA-A or HLA-B alleles of the respective melanoma cell lines. We then selected those 1000 peptides for each melanoma cell line that had the highest %-ranks for any of the alleles of the respective patient's haplotype. Sample HLA-typing information can be found in Supplementary Table 2.

For the melanoma tissues Ti1, Ti2, and TI3, the 500 top neoantigens as prioritized by NeoDisc were selected as targets. Additionally, we added the 518 TAA-derived class I peptides that were predicted by NeoDisc in the "ExpressedTAAs" table. Sample HLA-typing information can be found in Supplementary Table 2.

To efficiently schedule inclusion list entries and improve target detection, retention time prediction for target peptides is recommended. This approach minimizes the need for targeted scans to search for all peptides across the entire gradient, thereby reducing the loss in global coverage. For retention time prediction, the following are required: (1) retention times of calibration peptides, (2) a list of target peptides, and (3) a retention time prediction tool, such as DeepLC[56]. We used calibration peptide retention times derived from PSMs from a preceding tryptic HeLa digest (Thermo Fisher Scientific, ref. 88329) DDA measurement with the same LC gradient as the NeoDiscMS method. The calibration run should ideally be conducted shortly before the NeoDiscMS measurements and on the same column, ensuring multiple washed between them to minimize carry over. A spectral library was generated from the calibration run with FragPipe (v22.0) with default settings for tryptic digests, including methionine oxidation as a variable modification. Next, we predicted the RT of our list of targets with DeepLC (v3.1.1), with help of the RT peptide calibration file. Importantly, we included all methionine oxidation variations of target peptide sequences. To schedule our inclusion list, we created windows of ±15 min around the RT of each modified peptide (Supplementary Fig. 3a, b). Last, we generated the inclusion lists in the format dictated by manufacturer's method development software Thermo Xcalibur Instrument Setup (v2.0), including peptides with charge states 1–3.

For the RTS, we created dedicated FASTA files that consisted individual target peptide sequences as entries. For the melanoma cell line and the clinical tissue experiments, RTS FASTA headers were annotated as neoantigen or TAA. An example of sample Ml3 for the DeepLC calibration peptides file, DeepLC target peptides file, scheduled inclusion list, as well as a FASTA needed for RTS are provided as Supplementary Data 3–6.

An example acquisition tree, as seen in Thermo Xcalibur Instrument Setup, the order of adding the acquisition tree parameters with the drag-and-drop buttons, as well as where to find real-time search parameters is illustrated in Supplementary Fig. 1.

For all NeoDiscMS experiments, RTS parameters were set as visualized in Supplementary Fig. 1. Across experiments, settings remained the same apart from the inclusion list and FASTA used. Importantly, at the time of writing, all methods that apply RTS required a digestion enzyme as an input. Nevertheless, the database search then takes all digested peptides as well as the full FASTA sequences into account. Since the FASTA entries directly reflected the peptide sequences, we aimed to minimize forced digestion. In addition, no static modifications were allowed, and methionine oxidation was set as the only variable modification. We allowed up to three variable modifications/peptide. Maximum search time was set to 40 ms. To pass RTSf, the scoring thresholds were set to Xcorr -> 0.4, dCn -> 0, Precursor PPM -> 5 for charge states 1–3.

For all DIA experiments, we applied an MS1 mass range of 300–1650Th at a resolution of 120,000 and 27 DIA windows of sizes [37, 30, 24, 24, 22, 23, 44, 21, 24, 24, 25, 27, 27, 30, 35, 38, 43, 53, 72, 103, 594] with an AGC target of 2000%, the injection time set to auto, MS2 mass resolution set to 30,000, and a stepped CE of [27, 30, 32].

The injection volumes of each sample as well as the respective sample input are indicated in Supplementary Data 1. DDA, ilDDA, and NeoDiscMS acquisition cycles were constructed as described in Fig. 1b. Cycle times for all methods are 3 s. Detailed scan type parameters for all scan types are described in Supplementary Table 3. For ilDDA and NeoDiscMS, the according inclusion list/FASTA was included in the method.

We used an Easy-nLC 1200 coupled to an Orbitrap Eclipse Tribrid mass spectrometer (Thermo Fisher Scientific). Our analytical columns were 450 mm long with an inner diameter of 75 μm and an 8 μm tip (TSP-075375, BGB Analytik). For column packing we used C18 beads with a particle size of 1.9 μm and a pore size of 120 angstrom (r119.aq, Dr. Maisch). For all experiments, LC gradients used 0.1% formic acid (Thermo Scientific, prod nr. 85178) as solvent A and 80% ACN, 0.1% formic acid as solvent B, a gradient length of 125 min, and a flow rate of 250 nL/min. The gradient consisted of multiple, linear increases of solvent B: 0–110 min with 2–25%, 110–114 min with 25–35%, 114–115 min with 35–100%, 115–125 min with 100%.

To generate a calibration run for the DeepLC RT prediction, we injected 100 ng of tryptic HeLa digest that was measured with the same gradient above, with a MSFragger-DDA database search with all human Swissprot[57] sequences without isoforms (28.3.2024, 20419 entries) as well as the "common contaminants" list provided by FragPipe. We kept all default setting for tryptic searches in place.

### Immunopeptidomics searches with MSFragger-DDA

For immunopeptidomics searches with MSFragger-DDA, we applied a DDA database search with all human Swissprot sequences without isoforms (28.3.2024, 20,419 entries) as well as iRT peptide sequences and the "common contaminants" list provided by Fragpipe as a FASTA database. Digestion was set to unspecific, peptides of lengths 8–14 were included. Methionine oxidation, N-terminal acetylation and cysteine carbamidomethylation were set as variable modifications. Only precursors with charges 1–3 were permitted for PSMing. Protein FDR was set to 1. Peptide, ion, and psm FDRs were set to 0.01. For immunopeptidomics searches with MSFragger-DDA+ we applied DDA+ database searches with different FASTA databases for different experiments, though they were all concatenated with iRT peptide sequences and the "common contaminants" list provided by Fragpipe. For the JY- and RA957-based experiments we used all human Swissprot sequences without isoforms (28.3.2024, 20,419 entries). For melanoma cell lines and melanoma tissues the respective sample-specific FASTA generated by NeoDisc were used. Digestion was set to unspecific, peptides of lengths 8–14 were included. Methionine oxidation, N-terminal acetylation and cysteine carbamidomethylation were set as variable modifications. Only precursors with charges 1–3 were permitted for PSMing. DDA+-specific parameters were left at default. Protein FDR was set to 1. Peptide, ion, and psm FDR was set to 0.01 for with the SwissProt FASTA or 0.03 with group-specificity[52,58] for searches with NeoDisc FASTAs, respectively.

For elution profile visualization, raw files and target precursor sequence were imported into Skyline v22.2[59] for visualization.

For the spectrum-centric search with DIA-Umpire of DIA data from the melanoma tissue experiment, we did a database search with DIA-Umpire using a NeoDisc-generated FASTA. Digestion was set to unspecific, peptides of lengths 8–14 were included. Methionine oxidation, N-terminal acetylation and cysteine carbamidomethylation were set as variable modifications. DIA-Umpire-specific parameters were left at default. Only precursors with charges 1–3 were permitted for PSMing. Protein FDR was set to 1. Peptide, ion, and PSM FDR was set to 0.03 with group-specificity.

From the libraries generated with a spectrum-centric search of the respective DIA data and the NeoDiscMS MSFragger-DDA+ results of either Ti1, Ti2, or Ti3, we generated a hybrid library for each tissue separately. We used said hybrid library, per tissue, for a peptide-centric search of the DIA data with DIA-NN. FDR was set to 0.01.

## Data analysis and visualization

For immunopeptidomics data analysis we extracted psm.tsv output files of all spectrum-centric searches and report.tsv of all peptide-centric searches for data analysis. For peptide binding affinity prediction we used MixMHCpred[36] (v2.3). Data wrangling as well as analysis was done in R and Julia. Raw file header information was extracted in R with help of the rawrr[60] package (v1.12.0). To calculate precursor intensities per raw file we extracted the highest precursor PSM intensity for each precursor separately. We calculated peptide intensities by adding up all precursor intensities of a given peptide sequence. Data visualization was done with R in Rstudio with packages ggplot2[61] (v3.5.1), ggseqlogo[62] (v0.2), ggside (v0.3.1), patchwork (v1.3.0), ggh4x (v0.2.8), gt (v0.11.0).

## Implementation of NeoDiscMS

NeoDiscMS was introduced in NeoDisc v1.7.1 and can be executed through three steps: (1) Run NeoDisc to identify and prioritize immunogenic tumor-specific antigens from NGS data and predict their chromatographic RTs, (2) perform NeoDiscMS-based measurements of the immunopeptidome, (3) resume the NeoDisc workflow to search NeoDiscMS immunopeptidomic files against personalized references, followed by prioritization of tumor-specific antigens.

By running NeoDisc in fastq mode with the -neodiscmsselect flag, the user has to provide and specify in the configuration file a MS raw file of either shotgun proteomics or immunopeptidomics, as NEODISCMS_CALIBRATION parameter, that will be searched by FragPipe v22.0 for creating a calibration library used for RT prediction. This search is done against a GENCODE v43 protein-coding sequence reference, accounting for peptide modifications and the maximum number of modifications per peptide defined in the configuration file (MS_MODIFICATIONS and MS_MAXMODSPERPEPTIDE parameters). NeoDisc then partitions the prioritized HLA-I- and HLA-II-restricted peptide sequences into neoantigens, TAAs, and viral peptides, with the number of peptides in each group, as determined by the user in the NeoDisc configuration file (NEODISCMS_NEOCI, NEODISCMS_TAACI, NEODISCMS_VIRCI, NEODISCMS_NEOCII, NEODISCMS_TAACII, NEODISCMS_VIRCII). Their retention times is then predicted by DeepLC[56] v2.2.36, where the above mentioned calibration run library (peptide identification list with measured retention times) is used, and by considering peptide modifications (MS_MODIFICATIONS and MS_MAXMODSPERPEPTIDE), maximum charges per peptide (NEODISCMS_CHARGES), collision energy (NEODISCMS_CE), gradient length (NEODISCMS_GRADIENT), and the retention-time window width (NEODISCMS_WINRT) as defined in the NeoDisc configuration file. NeoDisc generates separate lists of HLA-I and HLA-II target peptide sequences (fasta) along with their predicted retention times (csv) specifically designed for NeoDiscMS measurements. After completing the NeoDiscMS measurements, the resulting raw files are added to the configuration file under MS_SPECTRA_SAMPLES, MS_SPECTRA_HLA_I, and MS_SPECTRA_HLA_II. The NeoDisc analysis is then resumed in fastq mode using the -neodiscms flag. This step involves searching the NeoDiscMS immunopeptidomic raw files with both Comet v2024.02_0 - NewAnce v1.7.5 and Fragpipe v22.0 DDA+ against the personalized references generated by NeoDisc. The results are subsequently integrated into the final NeoDisc analysis and prioritization.

## Statistics and reproducibility

To ensure reproducibility of results we performed all comparisons of different conditions in triplicates, as commonly used in the field. No data was excluded. To investigate different technical conditions, such as different acquisition parameters, measurements were randomized.

## Reporting summary

Further information on research design is available in the Nature Portfolio Reporting Summary linked to this article.

## Data availability

Raw files and real-time search tables generated during the acquisition, as well as identified PSMs of NeoDiscMS measurement, have been deposited to the ProteomeXchange Consortium via the PRIDE[63] partner repository with the dataset identifier PXD059824. Immunogenicity assessments and NGS data for Ml1 and Ml2 were described in Müller et al.[7], where they equate to patient1 and patient3, respectively. NGS and immunogenicity assessments for Ml3 is reported in Huber et al.[9], corresponds to Mel-4 P1 HLA-I high cells. WES and RNAseq data for TI1, Ti2, and Ti3 is deposited on the European Genome-Phenome Archive (EGA) with the dataset identifier EGAD50000001422.

## Code availability

NeoDiscMS is integrated within NeoDisc v1.8.0 and is available at https://neodisc.unil.ch/.

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

## Acknowledgements

We thank the staff of the CTE Biobank and of the Department of Oncology at the CHUV for their assistance. We are thankful to Marie Taillandier Coindard, Laura Wessling, and HuiSong Pak for their support, as well as to all other members of the immunopeptidomics lab at the Department of Oncology UNIL CHUV for their contributions to the discussion of the manuscript. This study was supported by the Ludwig Institute for Cancer Research, by grant KFS-5637-08-2022 from the Swiss Cancer Research Foundation (M.B.-S.) and the PRIMA grant PR00P3_193079 from the Swiss National Science Foundation (M.B.-S.).

## Author contributions

I.E.S. and M.B.-S. initiated the project, interpreted the results, and wrote the manuscript with contributions from all authors. I.E.S. performed the immunopeptidomics experiments including sample preparation, method implementation, data acquisition, and bioinformatical follow-up (data processing, analysis, and visualization). J.M. performed the NGS experiments (WES and RNAseq). F.H. processed the NGS data and implemented a module in the NeoDisc pipeline to provide users with the required inputs to apply NeoDiscMS and contributed the respective section in the methods. M.B.-S. provided study material and funding. All authors contributed valuable feedback for the improvement of the manuscript.

## Competing interests

The authors declare no competing interests.
