## [Transparent Peer Review file · Nature Communications]

Sensitive neoantigen discovery by real-time mutanome-guided immunopeptidomics

Corresponding Author: Professor Michal Bassani-Sternberg

Version 0:

Reviewer comments:

Reviewer #1

(Remarks to the Author)

Sensitive neoantigen discovery by real-time mutanome-guided immunopeptidomics

The current manuscript by Ilja Shapiro et al. describes NeoDiscMS, a novel mass spectrometric methodology exemplified on an Orbitrap Eclipse Tribrid mass spectrometer for simultaneous targeted and untargeted detection of immunopeptides. The authors describe and benchmark the method extensively using a variety of different cell line models and tumor samples showcasing the advantages of NeoDiscMS over classical DDA-based and inclusion list-based approaches. NeoDiscMS, similarly to previous approaches, utilizes an inclusion list to target a pre-defined list of neoepitopes or epitopes from TAAs. In contrast, the new method however applies a short scouting MS2 scan first for real time searching before dedicating to long, high sensitivity MS2 scans. The method in parallel also features wide window data dependent MS2 scans for untargeted immunopeptidome detection. This offers high sensitivity, to detect the most interesting potentially presented neoepitopes, while at the same time still generates a comprehensive picture of the global immunopeptidome to understand potential defects in the antigen processing and presentation machinery of the respective tumor or cell line, which also represents crucial information for epitope prioritization.

The manuscript is very well written and NeoDiscMS clearly offers improved detection capabilities particularly for cancer neoepitopes and TAAs. Before this work can be accepted in the prestigious Nature Communications journal though, the reviewer has a few suggestions, remarks and questions.

1. Abstract: Please define TAA before its first use.
2. Intro: Mass spectrometry (MS) was already defined in the abstract. So the abbreviation may be used already without the full term being stated.
3. Wide window MS2 isolation in DDA has been described recently by other groups for low input (PMID: 38310095) and single cell samples (PMID: 36802514 and 37380610) for classical proteomics. Please cite their works since wide window acquisition is also an integral part of the NeoDiscMS method.
4. Also concerning the wide isolation width used for MS2: what was the rational behind choosing exactly 3.2 Th as isolation width? Were there tests done before or any other rational considerations behind? From the aforementioned manuscripts it is clear that with increasing injection amounts (and hence sample complexity) the ideal window size becomes smaller. It would be great if this could be discussed in the manuscript.
5. Please comment on the observed peak width using your 125min method and the 45cm column. Even though precision and accuracy of quantification is not the most important feature in this workflow: does the 3s cycle time allow a decent number of points per peak and hence reproducible quantification? What are the peak widths you are observing?
6. The workflow is demonstrated on an Orbitrap Eclipse Tribrid instrument but which other instruments from Thermo will allow to utilize this method? This reviewer believes that it would be quite beneficial for the reader (and the widespread application of the method) to also briefly mention which other instruments will be able to employ this method.
7. End of page 5/beginning of page 6: The term non-isolated precursor is misleading since it was isolated, but just not the precursor triggering the MS2, if I understand this sentence correctly. I would modify this term to co-isolated. Can you discuss the fact that the deviations seen from the MS2-triggering precursor are integers/z? As it is written at the moment this is only stated as a fact but not discussed at all.
8. Immunopeptide enrichment in methods: DTT was probably not used for the crosslinking. I assume it should read DMP.
9. Figure 2b: While the figure does provide the intended information, this reviewer believes that the data could be presented in a more condensed way that is also quicker to grasp. Please consider changing the graph type.
10. Figure 4a: Add a small axis label on the very top left to indicate frequency of detection or similar

(Remarks on code availability)

Reviewer #2

(Remarks to the Author)

In this manuscript, Shapiro et al. describe a novel immunopeptidome data acquisition strategy that incorporates neoantigens inferred from external sources to enhance acquisition sensitivity while maintaining control over the total number of immunopeptidome scans. The authors demonstrate: (a) that a wider isolation window, when combined with MSFragger-DDA+, improves the handling of co-fragmented MS2 scans; (b) enhanced sensitivity in identifying clinically relevant neoantigens compared to conventional DDA, DDA with an inclusion list, and DIA approaches; (c) increased confidence in identified peptides relative to existing methods. I have the following comments for the authors to consider:

Majors:

[1] While I certainly appreciate the bottleneck in neoantigen identification in clinical settings and the efforts to address this challenge, I want to ensure I fully understand the innovation presented here. From my reading of the manuscript, it seems that the primary added utility of NeoDiscMS is the incorporation of a 'real-time search' step following scouting MS2. However, this step is executed by instrument-dependent software rather than NeoDiscMS itself, which primarily generates the inclusion list and FASTA file before resuming after data acquisition.

If my understanding is correct (please correct me if I am wrong), would it be more accurate to describe the contribution as advocating for the integration of real-time search into immunopeptidome analysis, rather than presenting NeoDiscMS as a novel pipeline or extension?

[2] Building on my previous point, while the authors mention in the discussion that this approach can be applied to virtually all MS instruments, there is no demonstration of its performance on platforms from other vendors, such as Bruker TIMS or SCIEX. Would the reported improvement in sensitivity necessarily extrapolate to these systems? It's perfectly fine if not, but I believe it would be valuable to clarify this, as it would help researchers using different platforms make informed decisions about selecting the most appropriate acquisition method.

[3] Another conceptual question I have is regarding the trade-off between immunopeptidome depth and neoantigen identification. While the authors argue that reducing global immunopeptidome depth by selecting the most promising sMS2 is beneficial, I wonder whether this is truly a major concern, especially given the compelling results in Fig. 5, where the added benefit of hMS2 in generating more fragment ions is clear. Since this ultimately leads to the identification of more neoantigens, which is the primary goal of the application, wouldn't it be worthwhile to prioritize a deeper immunopeptidome?

[4] I also want to challenge the purported benefit of using a wider window scan. Conceptually, I agree that it can help account for precursor ions deviating from their theoretical m/z . However, in practice, co-fragmentation will inevitably lead to more complex MS2 spectra, which is not ideal. While deconvolution methods and secondary peptide searches are designed to address this unavoidable challenge, intentionally introducing this issue seems somewhat counterintuitive. I believe this point warrants further clarification.

[5] Perhaps this is a naive question, but I would like to better understand why NeoDiscMS is expected to have greater sensitivity than DDA with an inclusion list. While I see that it reduces depth and increases the confidence of identified peptides, wouldn't using the full inclusion list, conceptually, be more effective in maximizing sensitivity?

[6] Could the authors expand on the description of the DIA section in both the results and methods sections? I am unclear on how the spectral library was generated and what is meant by a spectrum-centric and hybrid DIA analysis. Additionally, I am unsure how NeoDiscMS can be integrated with DIA and why it would conceptually outperform DIA. If the key advantages proposed for NeoDiscMS are wider isolation windows and the inclusion of peptides of interest, wouldn't DIA already be an ideal choice, albeit with the trade-off of increased complexity and deeper coverage?

Minors:

[1] The term "Tissue ID" is inconsistently formatted throughout the text, sometimes as "Ti" and other times as "TI", I suggest standardizing it for clarity and consistency.

[2] I suggest reconsidering the terms 'global depth' and 'global presentation patterns,' as their meaning is not immediately clear. It took me several reads to fully grasp their intended meaning. Perhaps a more precise alternative, such as 'total number of scans,' would improve clarity.

[3] The introduction, as well as parts of the results, seem to assume that readers have a deep background in immunopeptidomics and mass spectrometry. I suggest providing brief explanations of key concepts, such as resolution, dynamic exclusion, chimeric spectra deconvolution, to make the text more accessible to a broader audience.

(Remarks on code availability)

The author previously described NeoDisc as a comprehensive pipeline to establish personalized proteome from matched DNA (tumor and matched control), RNA and immunopeptidome data. NeoDiscMS has been integrated in the latest version, and the whole pipeline is available to download free of charge for academic usage in the designed website, with manual and licenses files available.

Version 1:

Reviewer comments:

Reviewer #1

(Remarks to the Author)

The authors have fully addressed all my comments and suggestions. Hence, I suggest the current version of the manuscript for publication in Nature Communications.

(Remarks on code availability)

Reviewer #2

(Remarks to the Author)

I want to thank the authors for thoroughly responding to my previous concerns and clarified each point in the revised manuscript, I believe all my concerns have been resolved.

(Remarks on code availability)

The provided software has a dedicated web portal illustrating the steps, and is integrated into the parental NeoDisc software suite. The author provided a detailed PDF file for each step to allow users to follow. I believe it meets the standards.

NeoDiscMS revision

We want to thank the reviewers for scrutinizing our proposed manuscript and providing valuable feedback to improve the quality and clarity of our work. We addressed all points of concern point-by-point below.

We would also like to mention that we noticed that we mentioned the fact that hMS2 scans use stepped collision energies, but omitted this information to supplemental table 1. We corrected this.

Also, because we received such helpful inputs that we incorporated into the manuscript, we reordered the discussion to best accommodate these changes regarding the transition between subject and reading flow.

Reviewer #1 (Remarks to the Author)

Sensitive neoantigen discovery by real-time mutanome-guided immunopeptidomics. The current manuscript by Ilja Shapiro et al. describes NeoDiscMS, a novel mass spectrometric methodology exemplified on an Orbitrap Eclipse Tribrid mass spectrometer for simultaneous targeted and untargeted detection of immunopeptides. The authors describe and benchmark the method extensively using a variety of different cell line models and tumor samples showcasing the advantages of NeoDiscMS over classical DDA-based and inclusion list-based approaches. NeoDiscMS, similarly to previous approaches, utilizes an inclusion list to target a pre-defined list of neoepitopes or epitopes from TAAs. In contrast, the new method however applies a short scouting MS2 scan first for real time searching before dedicating to long, high sensitivity MS2 scans. The method in parallel also features wide window data dependent MS2 scans for untargeted immunopeptidome detection. This offers high sensitivity, to detect the most interesting potentially presented neoepitopes, while at the same time still generates a comprehensive picture of the global immunopeptidome to understand potential defects in the antigen processing and presentation machinery of the respective tumor or cell line, which also represents crucial information for epitope prioritization.

The manuscript is very well written and NeoDiscMS clearly offers improved detection capabilities particularly for cancer neoepitopes and TAAs. Before this work can be accepted in the prestigious Nature Communications journal though, the reviewer has a few suggestions, remarks and questions.

1. Abstract: Please define TAA before its first use.

Thank you for your feedback. **Please find the according change in the abstract where we first spelled the full term out.**

2. Intro: Mass spectrometry (MS) was already defined in the abstract. So the abbreviation may be used already without the full term being stated.

Thank you for your feedback. Since the abstract and main text are often read independently, it is commonly recommended to define abbreviations in the main text even if they have already been introduced in the abstract. **For this reason, we have retained the definitions. We will, of course, follow the editors' guidance on this matter during the proof stage.**

3. Wide window MS2 isolation in DDA has been described recently by other groups for low input (PMID: 38310095) and single cell samples (PMID: 36802514 and 37380610) for classical proteomics. Please cite their works since wide window acquisition is also an integral part of the NeoDiscMS method.

Thank you for your feedback. **We added the references in the last paragraph of the introduction.**

“Inspired by previous studies that applied chimeric spectrum deconvolution to identify tryptic peptides from low-input and single-cell samples, we incorporated similar algorithms capable of resolving multiple precursors from chimeric spectra 31–35”.

4. Also concerning the wide isolation width used for MS2: what was the rationale behind choosing exactly 3.2 Th as isolation width? Were there tests done before or any other rational considerations behind? From the aforementioned manuscripts it is clear that with increasing injection amounts (and hence sample complexity) the ideal window size becomes smaller. It would be great if this could be discussed in the manuscript.

Thank you for your feedback. At the beginning of our method optimization we tested four different isolation windows (1.2Th , 2.2Th , 3.2Th , 4.2Th) with measurement that correspond to 10mio JY cells/injection. Identification quality, as assessed by binding affinity prediction, remains consistent across conditions; the only difference lies in the number of unique peptides identified. We selected 3.2Th because it consistently yielded the highest number of identifications. **We included this information in the revised manuscript at the beginning of the results and with a new supplementary figure 2.**

Suppl. Fig. 2: Comparison of different MS2 window sizes when processing with MSFragger-DDA+. Number of uniquely identified peptides for DDA acquisitions of 10 million JY cells equivalent / injection (n = 2).

“To benchmark the performance of wide isolation windows and chimeric spectra deconvolution on immunopeptidomics data, we first tested different isolation window sizes with DDA injections of the equivalent of 10 million JY cells per injection, analyzed with MSFragger-DDA+. 96% of the identified 8-14 mer peptides were predicted by MixMHCpred37 as binders (%-rank < 2) for each measurement, indicating that identification quality is consistent across conditions. Differences observed between isolation window widths pertained to the number of unique peptides identified. We selected 3.2Th because it most consistently yielded the highest number of identifications (Supplementary Fig. 2). This is in accordance with previous findings that optimal wide window size for increased depth with chimeric spectral deconvolution from low input samples remains narrow 35,38.”

5. Please comment on the observed peak width using your 125min method and the 45cm column. Even though precision and accuracy of quantification is not the most important feature in this workflow: does the 3s cycle time allow a decent number of points per peak and hence reproducible quantification? What are the peak widths you are observing?

Thank you for your feedback.

Quantification is calculated by FragPipe based on MS1 intensity (highest MS1 precursor intensity reported for a precursor across the run defines its level of abundance)^{1,2}. The precursor quantification therefore does not take into account the spectral information of multiple MS2 scans. At the peptide level, abundance is then inferred by summing up the abundances of all precursors. To demonstrate the reproducibility of the quantification with our setup (125min, 45cm, 3s cycle time, comparing 'NeoDiscMS' vs 'DDA' methods), we focused here on the 1/1024 dilution experiment measurements. We show that the variability between replicates is pretty much the same between DDA and NeoDiscMS and between non-target peptide identifications (gray, $R^2=0.96\%-0.97\%$) and target (red, $R^2=0.92\%-0.99\%$). We see slightly lower values for one NeoDiscMS comparison, though this is marginal considering the relatively lower number of target IDs. Importantly, we only considered PSMs derived from masses that were the targets of isolation of their respective MS2 scan, as those should be considered for MS2-based quantification.

We added a paragraph to the "Scalable enhanced sensitivity for targets" section in the results that addresses the question of reproducibility of MS1-based quantification (which corresponds to the search engine output). We included a new subplot in figure 3 (3g) that shows the intensity correlations between technical replicates.

Repeated fragmentation via the targeted branch may open future opportunities to implement more accurate, advanced MS2-based quantification methods with NeoDiscMS data. However, hMS2 scans use different ion accumulation and collision energy settings compared to dMS2 and sMS2 scans. As a result, they introduce variations in both absolute fragment ion abundances and fragment intensity ratios, as illustrated by the example of the target peptide EVILIDPFHK++. We mention this potential caveat in the revised manuscript.

We also included new supplementary figure 5 that shows the above example for MS2 ion sampling of EVILIDPFHK++ over its elution curve as well as the visualization of data points per peak and peak width we show above.

The number of points per peak and peak width are not directly provided by FragPipe. To answer this comment, we reprocessed the NeoDiscMS and DDA files of the 1/1024 and 1/16 dilution experiments with MaxQuant version 2.1 with similar search parameters and used the allPeptides.txt and msmsScan.txt files to derive peak width and data points per peak parameters. The plots below demonstrate the peak width in seconds of the identified targets and non-target HLA peptides. For the targets detected in the 1/1024 and 1/16 dilution samples, the median peak width is 25 seconds in DDA and 23 seconds in NeoDiscMS data. For all other HLA peptides, the median peak width is 28 seconds in DDA and 30 seconds in NeoDiscMS data.

Peak with of identified target peptides of 1/16 and 1/1024 dilution DDA and NeoDiscMS injections:

Peak with of non-target peptides of 1/16 and 1/1024 dilution DDA and NeoDiscMS injections:

Furthermore, we plotted the number of scans (the number of MS1 scans that the 3D peaks of a peptide feature) against retention length (the total retention time width of the peak) in seconds for all peptides. Majority of the peptides have more than ten MS1 scans and we conclude that a cycle of 3 seconds in our setup is sufficient to allow decent MS1 quantification.

However, to avoid any confusion regarding the use of MaxQuant solely for this specific purpose— as opposed to FragPipe, which was used consistently throughout the manuscript—we decided not to include the information obtained from MaxQuant in the revised version. We hope the reviewer agrees with this decision.

6. The workflow is demonstrated on an Orbitrap Eclipse Tribrid instrument but which other instruments from Thermo will allow to utilize this method? This reviewer believes that it would be quite beneficial for the reader (and the widespread application of the method) to also briefly mention which other instruments will be able to employ this method.

Thank you for your feedback. The instruments where NeoDiscMS can currently be set up with the method developer as provided by the manufacturer are Tribrid Eclipse and Tribrid Ascend instruments. **We have added this information to the manuscript for clarity.**

7. End of page 5/beginning of page 6: The term non-isolated precursor is misleading since it was isolated, but just not the precursor triggering the MS2, if I understand this sentence correctly. I would modify this term to co-isolated. Can you discuss the fact that the deviations seen from the MS2-triggering precursor are integers/z? As it is written at the moment this is only stated as a fact but not discussed at all.

Thank you for your feedback and for raising these concerns.

We now use the term “co-isolated” as suggested by the reviewer to improve clarity.

In a chimeric spectrum, the mass difference between the two peptides is defined by combinations of amino acids whose weight permits it to generate precursors that can be isolated through the same isolation window^{3,4}. Because the residuals of the amino acid weights all lie in a narrow 0.1Da range (see table below), we observed, as expected, that the absolute difference in mass (Da) between co-isolated peptides is often very close to multiples of 1 Da (when we filter for co-isolated identified peptides with the same charge).

To achieve the same precursor mass, specific single amino acid substitutions result in mass differences of around 1 Da (for example Glu and Gln, or Asn and Asp, see table below). If two peptides with very different sequences still appear to have the same precursor mass (m/z), then large mass differences introduced by substitutions must effectively cancel out (see example below), and the overall mass difference between substituted amino acids also tends to be close to multiples of 1 Da. Larger shifts in mass would otherwise result in a detectable difference in precursor m/z and these will not be co-isolated.

For example the two peptides identified in scan 9353:

dil	acq	TR	scan	precursorMass	psmed_precursor	psmed_observed_mz	psmed_calculated_mz	psmed_charge	psm_ppm_dev_from_isol_mass	psms_per_scan	isol_mass_psm	prec_weight
JY_only	DDA	1	9353	402.2020	KPNTSSKRSSL_3	402.2259	402.2263	3	890.8459025	2	chimering_psm	1206.6789
JY_only	DDA	1	9353	402.2020	RPSDANRKEM_3	401.8677	401.8681	3	0.2486313	2	isolmass_psm	1205.6043

The column «prec_weight» contains the calculated monoisotopic mass of each identified peptide. The difference between the two precursor weights in this example is:

$$1206.6789\text{Da} - 1205.6043\text{Da} = 1.0746\text{Da}$$

In the revised manuscript, we decided to demonstrate this by plotting the weight difference for all PSMs that were co-isolated for a single MS2 scan (New Fig.2f), and we added a shorter version of the explanation above.

If we now take the weight differences shown in the plot above and divide the mass difference by the charge (separately for charge 2 and charge 3) of the co-isolated identified peptides, we get the delta m/z distribution that we previously showed in fig 2f. We believe the new figure is more suitable to demonstrate this observation.

Former figure 2f:

Monoisotopic amino acid weights:

I-letter code	3-letter code	Chemical formula	Monoisotopic	Average
A	Ala	C ₃ H ₅ ON	71.03711	71.0788
R	Arg	C ₆ H ₁₂ ON ₄	156.10111	156.1875
N	Asn	C ₄ H ₆ O ₂ N ₂	114.04293	114.1038
D	Asp	C ₄ H ₅ O ₃ N	115.02694	115.0886
C	Cys	C ₃ H ₅ ONS	103.00919	103.1388
E	Glu	C ₅ H ₇ O ₃ N	129.04259	129.1155
Q	Gln	C ₅ H ₈ O ₂ N ₂	128.05858	128.1307
G	Gly	C ₂ H ₃ ON	57.02146	57.0519
H	His	C ₆ H ₇ ON ₃	137.05891	137.1411
I	Ile	C ₆ H ₁₁ ON	113.08406	113.1594
L	Leu	C ₆ H ₁₁ ON	113.08406	113.1594
K	Lys	C ₆ H ₁₂ ON ₂	128.09496	128.1741
M	Met	C ₅ H ₉ ONS	131.04049	131.1926
F	Phe	C ₉ H ₉ ON	147.06841	147.1766
P	Pro	C ₅ H ₇ ON	97.05276	97.1167
S	Ser	C ₃ H ₅ O ₂ N	87.03203	87.0782
T	Thr	C ₄ H ₇ O ₂ N	101.04768	101.1051
W	Trp	C ₁₁ H ₁₀ ON ₂	186.07931	186.2132
Y	Tyr	C ₉ H ₉ O ₂ N	163.06333	163.1760
V	Val	C ₅ H ₉ ON	99.06841	99.1326

8. Immunopeptide enrichment in methods: DTT was probably not used for the crosslinking. I assume it should read DMP.

Thank you for your feedback. **We corrected this in the revised manuscript.**

9. Figure 2b: While the figure does provide the intended information, this reviewer believes that the data could be presented in a more condensed way that is also quicker to grasp. Please consider changing the graph type.

Thank you for your feedback. We have averaged the values for each condition, and chose a new visualization that will hopefully make our data more clear to the reader. **The new plot in Figure 2 now also contains the absolute number of scans.**

10. Figure 4a: Add a small axis label on the very top left to indicate frequency of detection or similar

Thank you for your feedback. **We modified Fig. 4 as suggested.**

11. Respectfully is used a couple of times in the manuscript when it should read respectively. Please correct. (Remarks on code availability)

Thank you for bringing this to our attention. **We corrected this across the manuscript.**

Reviewer #3 (Remarks to the Author)

In this manuscript, Shapiro et al. describe a novel immunopeptidome data acquisition strategy that incorporates neoantigens inferred from external sources to enhance acquisition sensitivity while maintaining control over the total number of immunopeptidome scans. The authors demonstrate: (a) that a wider isolation window, when combined with MSFragger-DDA+, improves the handling of co-fragmented MS2 scans; (b) enhanced sensitivity in identifying clinically relevant neoantigens compared to conventional DDA, DDA with an inclusion list, and DIA approaches; (c) increased confidence in identified peptides relative to existing methods. I have the following comments for the authors to consider:

Majors:

[1] While I certainly appreciate the bottleneck in neoantigen identification in clinical settings and the efforts to address this challenge, I want to ensure I fully understand the innovation presented here. From my reading of the manuscript, it seems that the primary added utility of NeoDiscMS is the incorporation of a 'real-time search' step following scouting MS2. However, this step is executed by instrument-dependent software rather than NeoDiscMS itself, which primarily generates the inclusion list and FASTA file before resuming after data acquisition.

If my understanding is correct (please correct me if I am wrong), would it be more accurate to describe the contribution as advocating for the integration of real-time search into immunopeptidome analysis, rather than presenting NeoDiscMS as a novel pipeline or extension?

Thank you for your feedback. NeoDiscMS describes an LC-MS immunopeptidomics method for personalized clinical T-cell antigen discovery workflows. It is indeed not a pipeline. Since

NeoDiscMS extends the functionality of NeoDisc⁵ (list of prioritized targets have now an additional use: they enhance target detection when used with NeoDiscMS), we do consider it an extension of NeoDisc. It is NeoDisc that prioritizes targets and produces the inclusion and FASTA file that serve as inputs to the NeoDiscMS mass spectrometry acquisition, allowing to personalize the method for each (patient) sample.

The innovative aspects of NeoDiscMS are:

- Elevating the personalization of clinical immunopeptidomics workflow to the data acquisition level by incorporating NeoDisc-prioritized antigens into the method.
- Maximizing immunopeptide target sensitivity while minimizing loss of global depth by:
 - o splitting the method tree into two MS2 scan modules: discovery branch and targeted branch
 - o using scouting scans and subsequent RTS filter to selectively trigger high sensitivity scan system in the targeted branch. Additionally, no dynamic exclusion applies to provoke repeated fragmentation to further increase identification confidence.
 - o using wider isolation windows to benefit more from the deconvolution of chimeric spectra for more global depth^{3,4}.
- Does not rely on spike-ins or prior runs to define targets⁶⁻⁹.

None of these aspects have been previously reported for clinical immunopeptidomics or immunopeptidomics in general. NeoDiscMS combines these elements successfully to provide enhanced performance to our pipeline that needs to operate under specific limitations such as low sample input and short turnaround time. The reviewer is correct about the critical importance of real-time searches to improve the performance of NeoDiscMS.

We designed NeoDiscMS to be simple and affective making it easier for the community to adopt. To that end, NeoDiscMS integrates available tools and concepts (e.g. NeoDisc⁵ – somatic mutation calling and immunogenicity prediction, DeepLC¹⁰ – retention time prediction, RTS – real-time database search, MSFragger-DDA+³ - chimeric spectrum deconvolution) into a method that addresses previously unmet needs for clinical immunopeptidomics.

Inspired by the reviewer's feedback we added a comment in the introduction adding emphasis on the modularity of NeoDiscMS as this was not spelled out there as such yet but only in the discussion. The other points of innovation (spike-in free and scalable discovery with increased sensitivity, personalization through preceding NGS data processing, sMS2-RTSf-hMS2 system for selective triggering of high-sensitivity scans, use of wider isolation windows to increase depth with chimeric spectral deconvolutions) were already addressed in the introduction and discussion.

[2] Building on my previous point, while the authors mention in the discussion that this approach can be applied to virtually all MS instruments, there is no demonstration of its performance on platforms from other vendors, such as Bruker TIMS or SCIEX. Would the reported improvement in sensitivity necessarily extrapolate to these systems? It's perfectly fine if not, but I believe it would be valuable to clarify this, as it would help researchers using different platforms make informed decisions about selecting the most appropriate acquisition method.

Thank you for your feedback.

We assume that the reviewer refers to the statement “Importantly, NeoDiscMS is technically compatible with any MS architecture capable of generating MS/MS spectra.” towards the end of the discussion.

The architecture of mass spectrometers refers to the respective hardware (the ion source, mass analyzers, detectors, electronics, etc.) of an MS instrument. The MS architecture that is required to use NeoDiscMS is indeed very basic: the capability of generating MS/MS spectra and simple computations under communication with an external computer (as is also necessary for regular DDA methods to select peaks or maintain dynamic exclusion lists, for instance). This stands in contrast to other methods using real-time search that require an additional mass analyzer + detector to run, or the capability of generating MS³ spectra.

Currently, real-time search is available for the acquisition method editor software for Tribrid Eclipse and Tribrid Ascend instruments by Thermo. The reviewer is right—NeoDiscMS cannot be directly implemented on most instruments with vendor-provided method editors, due to software limitations. This, not because of constraints of the mass spectrometer architecture, but rather due to limitations of vendor-provided software solutions. We appreciate that the quoted statement could, hence, confuse readers, nonetheless. **We replaced the statement with just a description of which instruments let users implement NeoDiscMS at the time of writing.** We hope that this study will encourage broader vendor adoption of this approach and the integration of real-time search (RTS) capabilities.

“At the time of writing, the instruments equipped with vendor-supported acquisition method editors that enable implementation of NeoDiscMS are the Orbitrap Eclipse and Orbitrap Ascend (part of Thermo Fisher Scientific’s Tribrid Mass Spectrometer series). We envision a broader vendor adoption of this approach in the future and the integration of advanced RTS capabilities.”

[3] Another conceptual question I have is regarding the trade-off between immunopeptidome depth and neoantigen identification. While the authors argue that reducing global immunopeptidome depth by selecting the most promising sMS² is beneficial, I wonder whether this is truly a major concern, especially given the compelling results in Fig. 5, where the added benefit of hMS² in generating more fragment ions is clear. Since this ultimately leads to the identification of more neoantigens, which is the primary goal of the application, wouldn’t it be worthwhile to prioritize a deeper immunopeptidome?

Thank you for your feedback. The reviewer is correct that identifying neoantigens and other targets is a key goal of NeoDiscMS. The clinical value of confidently identifying targets such as neoantigens is clear^{5,11}. Yet, in our view, this does not render global depth irrelevant. Most importantly, clinical tumor samples are obtained through highly invasive procedures from patients coping with severe burden of illness and treatment. These individuals, along with their families, willingly contribute precious tissue samples. Out of respect for their contribution, we view it as an ethical obligation to extract the greatest possible value from these specimens. Given the demonstrated benefits of comprehensive data depth (see below), it would be irresponsible to forgo this opportunity and risk losing critical information—both for immediate therapeutic design and for advancing future research. We believe it is crucial for both our team and the broader research community to develop and implement methods that optimize the insights derived from these invaluable clinical samples.

The importance of global depth in immunopeptidomics, that informs on the global immunopeptidome of the tumor sample, has several reasons:

- As we show in suppl. Fig. 2f, the discovery branch also provides sensitivity for targets. Global depth directly supports target discovery.
- The discovery branch might identify relevant peptides that were not prioritized, such as canonical and non-canonical TAAs. This can be important for studies where new classes of antigens/different protein databases are used for research.
- Immunopeptidomics data serves as a window into the dynamics of antigen processing¹²⁻¹⁴. The immunopeptidome of tumor samples is actively studied to better understand antigen processing in tumors, which is a vulnerable pathway that is frequently altered by cancer for immune evasion^{15,16}. Immunopeptidomics data from primary tumor samples is the most valuable in this respect, as it enables the investigation of clinically relevant, real-world scenarios.
- Global depth also provides further ways to retrospectively analyze clinical cohorts to extract knowledge, such as understanding the role of PTMs in adaptive immunity, for instance.
- Data of the global immunopeptidome directly contributes to improve immunogenicity predictions. This can be done by referring to entire databases, or to investigating the immunopeptidome of a specific sample for personalized immunotherapy design^{5,17-20}.
- On a purely bioinformatical level, one of the signature advantages of NeoDiscMS is that it can be searched in a “high mass resolution DDA”-like setting with search engines. This means that the discovery branch will provide a lot of unbiased MS2 spectra that are PSMed allowing for target-decoy FDR calculations. Having a high number of unbiased PSMs facilitates:
 - better FDR control due to a large variety and absolute number of PSMed peptides without the bias of sampling for particular peptides, as by our prioritization. This facilitates more robust statistics²¹⁻²³.
 - better use of rescoring algorithms, which depend on extracting features of PSM-candidate peptides to optimize score calculation for the separation of target and decoy populations. The DDA branch is guaranteed to provide spectra from abundant peptides which have a higher chance of showing high-quality MS2 spectra where features can be learned from. Rescoring brings particular benefit to immunopeptidomics data, making this point especially noteworthy²⁴.

We thank the reviewer for bringing up the question of global depth that made us realize that it was necessary to better introduce these aspects in the manuscript. **We have, therefore, dedicated an entire paragraph to highlight the value of the discovery branch in the discussion to address the importance of global depth in clinical immunopeptidomics.**

[4] I also want to challenge the purported benefit of using a wider window scan. Conceptually, I agree that it can help account for precursor ions deviating from their theoretical m/z. However, in practice, co-fragmentation will inevitably lead to more complex MS2 spectra, which is not ideal. While deconvolution methods and secondary peptide searches are designed to address this unavoidable challenge, intentionally introducing this issue seems somewhat counterintuitive. I believe this point warrants further clarification.

Thank you for your feedback.

One key metric to benchmark a processing method for identification in LC-MS-based proteomics (which includes immunopeptidomics) is the ability to provide FDR controlled identifications of high quality^{21,22,25}. As shown in fig. 2, there is added value to using wider isolation windows when aiming to deconvolute chimeric spectra as we identify more unique, high quality immunopeptides. The quality metrics that we use to assess these identifications are binding affinity predictions and peptide length distribution, both of which are the two key parameters to assess data quality in immunopeptidomics, and both of which are orthogonal to the scoring by the search engine (as long as the search engine is agnostic to HLA restriction). The quality of the set of peptides identified with our wide-window MSFragger-DDA+ strategy is equal to the quality of peptides that we identify when we use regular DDA isolation window sized. It is worth noting that the wide window isolation strategy impacts only the discovery branch, and in addition, users can implement NeoDiscMS without enabling the wide window isolation strategy, should they choose to do so.

We appreciate the concern that more complex MS2 spectra are problematic when deconvoluting chimeric spectra. We would kindly ask the reviewer to reconsider his/her concerns, pointing to the publications that describe algorithms for chimeric spectrum deconvolution^{3,4} and why they work so well. We hope we could convince the reviewer with our benchmarking metrics that there is added benefit in our approach and chimeric spectrum deconvolution, confirming the findings the authors of the indicated algorithms provide.

[5] Perhaps this is a naive question, but I would like to better understand why NeoDiscMS is expected to have greater sensitivity than DDA with an inclusion list. While I see that it reduces depth and increases the confidence of identified peptides, wouldn't using the full inclusion list, conceptually, be more effective in maximizing sensitivity?

Thank you for your feedback and questions regarding the performance of NeoDiscMS compared to iLDDA.

Typically, when using inclusion lists, the number of peptide targets is in the 10s or 100s, not reaching the NeoDiscMS benchmarked scale of 1'500. There are examples of workflows that focus on just targeting (like PRM), however, these workflows disregard global depth which is a requirement to us. In general, the performance of NeoDiscMS compared to iLDDA regarding target sensitivity is equal or slightly better. The reason for the possibility to perform better than iLDDA is the fact that the absolute number and features of scans on target precursors in NeoDiscMS is slightly different from an equivalent iLDDA measurement. Several factors play a role in this regard:

- The proportion of precursors captured by the inclusion list that are actually the target precursor is very small. Using a high-sensitivity scan on a precursor that is not the target but an isobaric molecule does not enhance target sensitivity and wastes time that could be better allocated to the discovery branch. The scouting scans, combined with real-time search filters, effectively manage this scenario.
- High-sensitivity scans without any selectivity take up more absolute time than first assessing the potential of a precursor with a scouting scan. The time that NeoDiscMS gains by being selective about high-sensitivity scans can be spent on further MS2 scans of eluants. Each of these scans bears a chance to identify a target as well.
- When the targeted branch is activated in NeoDiscMS up to two MS2 scans are triggered, depending on the real-time search filters. For iLDDA it is always only one scan. As each scan bears the chance to get PSMed to the target peptide, we have either the same or

better chances for this with NeoDiscMS compared to iLDDA. In cases where both iLDDA and NeoDiscMS only perform one scan with the targeted branch, the real-time search yielded a bad result. This indicates that the identification of that precursor as the target peptide is very unlikely to begin with.

We added a comment in the results pointing to the fact that more than 85% of inclusion list hits turn out to be isobaric non-targets, as by the results of RTSf. We also expanded the discussion that explains why NeoDiscMS doesn't only suffer less from loss in global depth than iLDDA, but also why we see equal or slightly greater target sensitivity.

“More than 85% of the inclusion list hits during a measurement do not pass RTSf, indicating that they are isobaric non-targets (Supplementary Fig. 3c).”

We also added a comment in the section “Scalable enhanced sensitivity for targets” that quantifies the improved trade-off between loss in global depth and improved target sensitivity.

“With respect to the number of uniquely identified peptides, we note that on average, for every 1% of loss in global depth compared to DDA, we gain 2% in target sensitivity with iLDDA and 5.6% with NeoDiscMS. This difference in sensitivity trade-off quantifies the minimization of loss in global depth with help of RTSf very clearly.”

And in the discussion: “The edge of NeoDiscMS over iLDDA lies in more efficient application of high-sensitivity scans. Long injection times on every analyte that is isobaric to a target within the scheduled RT window becomes increasingly troublesome as inclusion lists are scaled: the number of targets is critical for inclusion list burden. NeoDiscMS significantly reduces time spent sampling off-target ions. This leaves more time within an acquisition cycle to sample ions of other target masses or with the discovery branch. An additional advantage of NeoDiscMS is of its ability to sample the same precursor up to three times per cycle, compared to a maximum of two times in iLDDA. Each MS2 scan contributes a spectrum that can yield a high-quality PSM. This increased sampling frequency explains the higher global depth and the equal or slightly improved target sensitivity observed with NeoDiscMS relative to iLDDA.”

[6] Could the authors expand on the description of the DIA section in both the results and methods sections? I am unclear on how the spectral library was generated and what is meant by a spectrum-centric and hybrid DIA analysis. Additionally, I am unsure how NeoDiscMS can be integrated with DIA and why it would conceptually outperform DIA. If the key advantages proposed for NeoDiscMS are wider isolation windows and the inclusion of peptides of interest, wouldn't DIA already be an ideal choice, albeit with the trade-off of increased complexity and deeper coverage?

Thank you for your feedback. We hereby clarify our descriptions and elaborate on the advantages of NeoDiscMS over DIA for clinical target discovery.

In a peptide-centric DIA analysis, a library of immunopeptides capturing features like retention times and fragmentation patterns (usually a sample-specific library generated from DDA data) is used to match DIA data ²⁶.

In a spectrum-centric DIA analysis, the DIA MS2 spectra is interpreted directly, without relying on MS2 spectra library. The complex DIA spectra are deconvoluted and searched against a peptide database similar to DDA ²⁶.

The principles of spectrum- and peptide-centric data processing methods are well established in the literature ²⁶⁻²⁹.

The “hybrid” term in “hybrid library (search)” is a reference to the fact that multiple types of data (in our case NeoDiscMS and spectrum-centric search of DIA, for other projects it is often DDA and DIA) are used together to generate a hybrid library, which is then used to search DIA data in a peptide-centric fashion ^{30,31}.

We define this use of ‘hybrid library’ and ‘hybrid library search’ when the term is used for the first time in the section “Direct clinical application” in the results.

We would like to clarify that NeoDiscMS does not generally outperform DIA. These two acquisition methods are substantially different and we therefore propose to combine the two, just like DDA and DIA are often combined to provide a library for deep peptide-centric DIA searches. We demonstrated the performance of NeoDiscMS and DIA individually, and in combination:

We measured the same immunopeptidomics sample with DIA and NeoDiscMS. The DIA file that was analyzed by itself in a spectrum-centric fashion yielded a relatively low number of peptide identifications, compared to the same sample measured with NeoDiscMS. This is visualized in Fig. 6a, where NeoDiscMS yielded around double uniquely identified peptides by FragPipe compared to DIA with a spectrum-centric search (DIA-NN). As such, NeoDiscMS outperforms DIA when processed with a spectrum-centric search in terms of depth, and also in terms of target detection sensitivity, as can be seen in Fig. 6c. **We added a comment to the first paragraph of the sub-chapter “Direct clinical application” in the results that highlight this point.**

DIA is an attractive method achieving great depth and sensitivity when provided with a deep library to perform a peptide-centric search ³². NeoDiscMS is therefore capable of generating sample-specific deep library that contains a high number high-confidence MS2 of our potential targets. **We added a comment to section “Direct clinical application” to highlight this point.**

An alternative to experimentally-derived libraries are computationally predicted libraries. This approach holds great promise, yet it’s performance in immunopeptidomics compared to empirical libraries is situational ³²⁻³⁴ and accurate FDR calculations might face processing-specific challenges that need to be evaluated, just as is the case for spectrum-centric searches ^{23,25}. **We added a comment to the discussion to highlight this point.**

Another important reason to not solely rely on DIA-derived peptide identifications (regardless of spectrum-centric or peptide-centric (with empirical or predicted libraries) processing) is the extreme complexity of DIA MS2 spectra. As already discussed above, when we want to identify neoantigens for patient interventions, we want to be as confident as possible about associating a spectrum with a peptide. For this we often resort to considering how spectra were processed and identified, and we also manually and visually inspect the spectra annotation. DIA MS2 spectrum annotation where precursors were isolated with a window of 20Th is much more ambiguous than an MS spectrum that was identified with an isolation window that is an order of magnitude smaller, because it will contain significantly less co-eluting precursor information that could lead

to random matching and skew fragment intensities. Having experimental evidence for a narrow-window MS2 is highly conducive to gain confidence in a PSM. If developments in the field of proteomics such as narrow-window DIA³⁵ might shift this balance remains to be seen. Therefore, we advocate for combining NeoDiscMS with DIA, to benefit as much from these complementary methods.

We added a comment to the discussion that highlights.

Minors:

[1] The term “Tissue ID” is inconsistently formatted throughout the text, sometimes as “Ti” and other times as “TI”, I suggest standardizing it for clarity and consistency.

Thank you for your feedback. We did indeed find one location where we spelled out “TI3” and not “Ti3”. **We changed this accordingly.**

[2] I suggest reconsidering the terms 'global depth' and 'global presentation patterns,' as their meaning is not immediately clear. It took me several reads to fully grasp their intended meaning. Perhaps a more precise alternative, such as 'total number of scans,' would improve clarity.

Thank you for your feedback.

We agree that these terms can be confusing. **We hence substituted the term “global presentation patterns” with “hotspot features”^{19,36}, a term that is known from the immunopeptidomics literature.**

We also added what we mean by global depth with the first use in the abstract (“the entirety of the measurable immunopeptidome”).

[3] The introduction, as well as parts of the results, seem to assume that readers have a deep background in immunopeptidomics and mass spectrometry. I suggest providing brief explanations of key concepts, such as resolution, dynamic exclusion, chimeric spectra deconvolution, to make the text more accessible to a broader audience.

Thank you for your feedback. We would like to address each of those terms point by point in the manuscript. Please find the according changes as indicated for each keyword.

- Resolution: we use this term in the methods section to describe mass spectrum acquisition parameters. We clarified what resolution is meant by assuring that we refer to “mass resolution”.
- Dynamic exclusion: We added a comment that explains the term where we mention “dynamic exclusion” for the first time in the introduction.
- Chimeric spectra deconvolution: We added a comment that explains the term “chimeric spectra” and “chimeric spectra deconvolution” where we mention “chimeric spectra deconvolution” for the first time in the introduction.

1. Yu, F. *et al.* Fast Quantitative Analysis of timsTOF PASEF Data with MSFragger and IonQuant. *Molecular & Cellular Proteomics* **19**, 1575–1585 (2020).
2. Kong, A. T., Leprevost, F. V., Avtonomov, D. M., Mellacheruvu, D. & Nesvizhskii, A. I. MSFragger: ultrafast and comprehensive peptide identification in mass spectrometry-based proteomics. *Nat Methods* **14**, 513–520 (2017).
3. Yu, F., Deng, Y. & Nesvizhskii, A. I. MSFragger-DDA+ enhances peptide identification sensitivity with full isolation window search. *Nat Commun* **16**, 3329 (2025).
4. Frejno, M. *et al.* Unifying the analysis of bottom-up proteomics data with CHIMERYS. *Nat Methods* (2025) doi:10.1038/s41592-025-02663-w.
5. Huber, F. *et al.* A comprehensive proteogenomic pipeline for neoantigen discovery to advance personalized cancer immunotherapy. *Nat Biotechnol* (2024) doi:10.1038/s41587-024-02420-y.
6. Stopfer, L. E. *et al.* Absolute quantification of tumor antigens using embedded MHC-I isotopologue calibrants. *Proc. Natl. Acad. Sci. U.S.A.* **118**, e2111173118 (2021).
7. Stopfer, L. E., Mesfin, J. M., Joughin, B. A., Lauffenburger, D. A. & White, F. M. Multiplexed relative and absolute quantitative immunopeptidomics reveals MHC I repertoire alterations induced by CDK4/6 inhibition. *Nat Commun* **11**, 2760 (2020).
8. Leddy, O. *et al.* Validation and quantification of peptide antigens presented on MHCs using SureQuant. *Nat Protoc* (2024) doi:10.1038/s41596-024-01076-x.
9. Pollock, S. B. *et al.* Sensitive and Quantitative Detection of MHC-I Displayed Neoepitopes Using a Semiautomated Workflow and TOMAHAQ Mass Spectrometry. *Molecular & Cellular Proteomics* **20**, 100108 (2021).
10. Bouwmeester, R., Gabriels, R., Hulstaert, N., Martens, L. & Degroeve, S. DeepLC can predict retention times for peptides that carry as-yet unseen modifications. *Nat Methods* **18**, 1363–1369 (2021).

11. Lang, F., Schrörs, B., Löwer, M., Türeci, Ö. & Sahin, U. Identification of neoantigens for individualized therapeutic cancer vaccines. *Nat Rev Drug Discov* **21**, 261–282 (2022).
12. Shapiro, I. E. *et al.* Deleterious KOs in the HLA Class I Antigen Processing and Presentation Machinery Induce Distinct Changes in the Immunopeptidome. *Molecular & Cellular Proteomics* **24**, 100951 (2025).
13. McShan, A. C. *et al.* TAPBPR promotes antigen loading on MHC-I molecules using a peptide trap. *Nat Commun* **12**, 3174 (2021).
14. Marijt, K. A. *et al.* Identification of non-mutated neoantigens presented by TAP-deficient tumors. *Journal of Experimental Medicine* **215**, 2325–2337 (2018).
15. Galassi, C., Chan, T. A., Vitale, I. & Galluzzi, L. The hallmarks of cancer immune evasion. *Cancer Cell* **42**, 1825–1863 (2024).
16. Dunn, G. P., Bruce, A. T., Ikeda, H., Old, L. J. & Schreiber, R. D. Cancer immunoediting: from immunosurveillance to tumor escape. *Nat Immunol* **3**, 991–998 (2002).
17. Pyke, R. M. *et al.* Precision Neoantigen Discovery Using Large-Scale Immunopeptidomes and Composite Modeling of MHC Peptide Presentation. *Molecular & Cellular Proteomics* **22**, 100506 (2023).
18. Tokita, S. *et al.* Identification of immunogenic HLA class I and II neoantigens using surrogate immunopeptidomes. *Sci. Adv.* **10**, eado6491 (2024).
19. Müller, M., Gfeller, D., Coukos, G. & Bassani-Sternberg, M. ‘Hotspots’ of Antigen Presentation Revealed by Human Leukocyte Antigen Ligandomics for Neoantigen Prioritization. *Front. Immunol.* **8**, 1367 (2017).
20. Kraemer, A. I. *et al.* The immunopeptidome landscape associated with T cell infiltration, inflammation and immune editing in lung cancer. *Nat Cancer* **4**, 608–628 (2023).
21. Elias, J. E. & Gygi, S. P. Target-Decoy Search Strategy for Mass Spectrometry-Based Proteomics. in *Proteome Bioinformatics* (eds. Hubbard, S. J. & Jones, A. R.) vol. 604 55–71 (Humana Press, Totowa, NJ, 2010).

22. Gupta, N., Bandeira, N., Keich, U. & Pevzner, P. A. Target-Decoy Approach and False Discovery Rate: When Things May Go Wrong. *J. Am. Soc. Mass Spectrom.* **22**, 1111–1120 (2011).
23. Wen, B. *et al.* Assessment of false discovery rate control in tandem mass spectrometry analysis using entrapment. Preprint at <https://doi.org/10.1101/2024.06.01.596967> (2024).
24. Yang, K. L. *et al.* MSBooster: improving peptide identification rates using deep learning-based features. *Nat Commun* **14**, 4539 (2023).
25. Nesvizhskii, A. I. Proteogenomics: concepts, applications and computational strategies. *Nat Methods* **11**, 1114–1125 (2014).
26. Ting, Y. S. *et al.* Peptide-Centric Proteome Analysis: An Alternative Strategy for the Analysis of Tandem Mass Spectrometry Data. *Molecular & Cellular Proteomics* **14**, 2301–2307 (2015).
27. Tsou, C.-C. *et al.* DIA-Umpire: comprehensive computational framework for data-independent acquisition proteomics. *Nat Methods* **12**, 258–264 (2015).
28. Li, K., Teo, G. C., Yang, K. L., Yu, F. & Nesvizhskii, A. I. diaTracer enables spectrum-centric analysis of diaPASEF proteomics data. *Nat Commun* **16**, 95 (2025).
29. Guo, T. & Aebersold, R. Recent advances of data-independent acquisition mass spectrometry-based proteomics. *Proteomics* **23**, 2200011 (2023).
30. Lapcik, P. *et al.* A hybrid DDA/DIA-PASEF based assay library for a deep proteotyping of triple-negative breast cancer. *Sci Data* **11**, 794 (2024).
31. Willems, P., Fels, U., Staes, A., Gevaert, K. & Van Damme, P. Use of Hybrid Data-Dependent and -Independent Acquisition Spectral Libraries Empowers Dual-Proteome Profiling. *J. Proteome Res.* **20**, 1165–1177 (2021).
32. Pak, H. *et al.* Sensitive Immunopeptidomics by Leveraging Available Large-Scale Multi-HLA Spectral Libraries, Data-Independent Acquisition, and MS/MS Prediction. *Molecular & Cellular Proteomics* **20**, 100080 (2021).

33. Wahle, M. *et al.* IMBAS-MS Discovers Organ-Specific HLA Peptide Patterns in Plasma. *Molecular & Cellular Proteomics* **23**, 100689 (2024).
34. Oliinyk, D. *et al.* diaPASEF analysis for HLA-I peptides enables quantification of common cancer neoantigens. *Molecular & Cellular Proteomics* 100938 (2025)
doi:10.1016/j.mcpro.2025.100938.
35. Guzman, U. H. *et al.* Ultra-fast label-free quantification and comprehensive proteome coverage with narrow-window data-independent acquisition. *Nat Biotechnol* **42**, 1855–1866 (2024).
36. Jappe, E. C., Kringelum, J., Trolle, T. & Nielsen, M. Predicted MHC peptide binding promiscuity explains MHC class I ‘hotspots’ of antigen presentation defined by mass spectrometry eluted ligand data. *Immunology* **154**, 407–417 (2018).

NeoDiscMS revision

We want to thank the reviewers for scrutinizing our proposed manuscript and providing valuable feedback to improve the quality and clarity of our work. We addressed all points of concern point-by-point below.

We would also like to mention that we noticed that we mentioned the fact that hMS2 scans use stepped collision energies, but omitted this information to supplemental table 1. We corrected this.

Also, because we received such helpful inputs that we incorporated into the manuscript, we reordered the discussion to best accommodate these changes regarding the transition between subject and reading flow.

Reviewer #1 (Remarks to the Author)

Sensitive neoantigen discovery by real-time mutanome-guided immunopeptidomics The current manuscript by Ilja Shapiro et al. describes NeoDiscMS, a novel mass spectrometric methodology exemplified on an Orbitrap Eclipse Tribrid mass spectrometer for simultaneous targeted and untargeted detection of immunopeptides. The authors describe and benchmark the method extensively using a variety of different cell line models and tumor samples showcasing the advantages of NeoDiscMS over classical DDA-based and inclusion list-based approaches. NeoDiscMS, similarly to previous approaches, utilizes an inclusion list to target a pre-defined list of neoepitopes or epitopes from TAAs. In contrast, the new method however applies a short scouting MS2 scan first for real time searching before dedicating to long, high sensitivity MS2 scans. The method in parallel also features wide window data dependent MS2 scans for untargeted immunopeptidome detection. This offers high sensitivity, to detect the most interesting potentially presented neoepitopes, while at the same time still generates a comprehensive picture of the global immunopeptidome to understand potential defects in the antigen processing and presentation machinery of the respective tumor or cell line, which also represents crucial information for epitope prioritization.

The manuscript is very well written and NeoDiscMS clearly offers improved detection capabilities particularly for cancer neoepitopes and TAAs. Before this work can be accepted in the prestigious Nature Communications journal though, the reviewer has a few suggestions, remarks and questions.

1. Abstract: Please define TAA before its first use.

Thank you for your feedback. **Please find the according change in the abstract where we first spelled the full term out.**

2. Intro: Mass spectrometry (MS) was already defined in the abstract. So the abbreviation may be used already without the full term being stated.

Thank you for your feedback. Since the abstract and main text are often read independently, it is commonly recommended to define abbreviations in the main text even if they have already been introduced in the abstract. **For this reason, we have retained the definitions. We will, of course, follow the editors' guidance on this matter during the proof stage.**

3. Wide window MS2 isolation in DDA has been described recently by other groups for low input (PMID: 38310095) and single cell samples (PMID: 36802514 and 37380610) for classical proteomics. Please cite their works since wide window acquisition is also an integral part of the NeoDiscMS method.

Thank you for your feedback. **We added the references in the last paragraph of the introduction.**

“Inspired by previous studies that applied chimeric spectrum deconvolution to identify tryptic peptides from low-input and single-cell samples, we incorporated similar algorithms capable of resolving multiple precursors from chimeric spectra 31–35”.

4. Also concerning the wide isolation width used for MS2: what was the rationale behind choosing exactly 3.2 Th as isolation width? Were there tests done before or any other rational considerations behind? From the aforementioned manuscripts it is clear that with increasing injection amounts (and hence sample complexity) the ideal window size becomes smaller. It would be great if this could be discussed in the manuscript.

Thank you for your feedback. At the beginning of our method optimization we tested four different isolation windows (1.2Th , 2.2Th , 3.2Th , 4.2Th) with measurement that correspond to 10mio JY cells/injection. Identification quality, as assessed by binding affinity prediction, remains consistent across conditions; the only difference lies in the number of unique peptides identified. We selected 3.2Th because it consistently yielded the highest number of identifications. **We included this information in the revised manuscript at the beginning of the results and with a new supplementary figure 2.**

Suppl. Fig. 2: Comparison of different MS2 window sizes when processing with MSFragger-DDA+. Number of uniquely identified peptides for DDA acquisitions of 10 million JY cells equivalent / injection (n = 2).

“To benchmark the performance of wide isolation windows and chimeric spectra deconvolution on immunopeptidomics data, we first tested different isolation window sizes with DDA injections of the equivalent of 10 million JY cells per injection, analyzed with MSFragger-DDA+. 96% of the identified 8-14 mer peptides were predicted by MixMHCpred37 as binders (%-rank < 2) for each measurement, indicating that identification quality is consistent across conditions. Differences observed between isolation window widths pertained to the number of unique peptides identified. We selected 3.2Th because it most consistently yielded the highest number of identifications (Supplementary Fig. 2). This is in accordance with previous findings that optimal wide window size for increased depth with chimeric spectral deconvolution from low input samples remains narrow 35,38.”

5. Please comment on the observed peak width using your 125min method and the 45cm column. Even though precision and accuracy of quantification is not the most important feature in this workflow: does the 3s cycle time allow a decent number of points per peak and hence reproducible quantification? What are the peak widths you are observing?

Thank you for your feedback.

Quantification is calculated by FragPipe based on MS1 intensity (highest MS1 precursor intensity reported for a precursor across the run defines its level of abundance)^{1,2}. The precursor quantification therefore does not take into account the spectral information of multiple MS2 scans. At the peptide level, abundance is then inferred by summing up the abundances of all precursors. To demonstrate the reproducibility of the quantification with our setup (125min, 45cm, 3s cycle time, comparing ‘NeoDiscMS’ vs ‘DDA’ methods), we focused here on the 1/1024 dilution experiment measurements. We show that the variability between replicates is pretty much the same between DDA and NeoDiscMS and between non-target peptide identifications (gray, $R^2=0.96\%-0.97\%$) and target (red, $R^2=0.92\%-0.99\%$). We see slightly lower values for one NeoDiscMS comparison, though this is marginal considering the relatively lower number of target IDs. Importantly, we only considered PSMs derived from masses that were the targets of isolation of their respective MS2 scan, as those should be considered for MS2-based quantification.

We added a paragraph to the “Scalable enhanced sensitivity for targets” section in the results that addresses the question of reproducibility of MS1-based quantification (which corresponds to the search engine output). We included a new subplot in figure 3 (3g) that shows the intensity correlations between technical replicates.

Repeated fragmentation via the targeted branch may open future opportunities to implement more accurate, advanced MS2-based quantification methods with NeoDiscMS data. However, hMS2 scans use different ion accumulation and collision energy settings compared to dMS2 and sMS2 scans. As a result, they introduce variations in both absolute fragment ion abundances and fragment intensity ratios, as illustrated by the example of the target peptide EVILIDPFHK++. We mention this potential caveat in the revised manuscript.

We also included new supplementary figure 5 that shows the above example for MS2 ion sampling of EVILIDPFHK++ over its elution curve as well as the visualization of data points per peak and peak width we show above.

The number of points per peak and peak width are not directly provided by FragPipe. To answer this comment, we reprocessed the NeoDiscMS and DDA files of the 1/1024 and 1/16 dilution experiments with MaxQuant version 2.1 with similar search parameters and used the allPeptides.txt and msmsScan.txt files to derive peak width and data points per peak parameters. The plots below demonstrate the peak width in seconds of the identified targets and non-target HLA peptides. For the targets detected in the 1/1024 and 1/16 dilution samples, the median peak width is 25 seconds in DDA and 23 seconds in NeoDiscMS data. For all other HLA peptides, the median peak width is 28 seconds in DDA and 30 seconds in NeoDiscMS data.

Peak width of identified target peptides of 1/16 and 1/1024 dilution DDA and NeoDiscMS injections:

Peak width of non-target peptides of 1/16 and 1/1024 dilution DDA and NeoDiscMS injections:

Furthermore, we plotted the number of scans (the number of MS1 scans that the 3D peaks of a peptide feature) against retention length (the total retention time width of the peak) in seconds for all peptides. Majority of the peptides have more than ten MS1 scans and we conclude that a cycle of 3 seconds in our setup is sufficient to allow decent MS1 quantification.

However, to avoid any confusion regarding the use of MaxQuant solely for this specific purpose— as opposed to FragPipe, which was used consistently throughout the manuscript—we decided not to include the information obtained from MaxQuant in the revised version. We hope the reviewer agrees with this decision.

6. The workflow is demonstrated on an Orbitrap Eclipse Tribrid instrument but which other instruments from Thermo will allow to utilize this method? This reviewer believes that it would be quite beneficial for the reader (and the widespread application of the method) to also briefly mention which other instruments will be able to employ this method.

Thank you for your feedback. The instruments where NeoDiscMS can currently be set up with the method developer as provided by the manufacturer are Tribrid Eclipse and Tribrid Ascend instruments. **We have added this information to the manuscript for clarity.**

7. End of page 5/beginning of page 6: The term non-isolated precursor is misleading since it was isolated, but just not the precursor triggering the MS2, if I understand this sentence correctly. I would modify this term to co-isolated. Can you discuss the fact that the deviations seen from the MS2-triggering precursor are integers/z? As it is written at the moment this is only stated as a fact but not discussed at all.

Thank you for your feedback and for raising these concerns.

We now use the term “co-isolated” as suggested by the reviewer to improve clarity.

In a chimeric spectrum, the mass difference between the two peptides is defined by combinations of amino acids whose weight permits it to generate precursors that can be isolated through the same isolation window^{3,4}. Because the residuals of the amino acid weights all lie in a narrow 0.1Da range (see table below), we observed, as expected, that the absolute difference in mass (Da) between co-isolated peptides is often very close to multiples of 1 Da (when we filter for co-isolated identified peptides with the same charge).

To achieve the same precursor mass, specific single amino acid substitutions result in mass differences of around 1 Da (for example Glu and Gln, or Asn and Asp, see table below). If two peptides with very different sequences still appear to have the same precursor mass (m/z), then large mass differences introduced by substitutions must effectively cancel out (see example below), and the overall mass difference between substituted amino acids also tends to be close to multiples of 1 Da. Larger shifts in mass would otherwise result in a detectable difference in precursor m/z and these will not be co-isolated.

For example the two peptides identified in scan 9353:

dil	acq	TR	scan	precursorMass	psmed_precursor	psmed_observed_mz	psmed_calculated_mz	psmed_charge	psm_ppm_dev_from_isol_mass	psms_per_scan	isol_mass_psm	prec_weight
JY_only	DDA	1	9353	402.2020	KPNTSSKRSSL ₃	402.2259	402.2263	3	890.8459025	2	chimering_psm	1206.6789
JY_only	DDA	1	9353	402.2020	RPSDANRKEML ₃	401.8677	401.8681	3	0.2486313	2	isolmass_psm	1205.6043

The column «prec_weight» contains the calculated monoisotopic mass of each identified peptide. The difference between the two precursor weights in this example is:

$$1206.6789\text{Da} - 1205.6043\text{Da} = 1.0746\text{Da}$$

In the revised manuscript, we decided to demonstrate this by plotting the weight difference for all PSMs that were co-isolated for a single MS2 scan (New Fig.2f), and we added a shorter version of the explanation above.

If we now take the weight differences shown in the plot above and divide the mass difference by the charge (separately for charge 2 and charge 3) of the co-isolated identified peptides, we get the delta m/z distribution that we previously showed in fig 2f. We believe the new figure is more suitable to demonstrate this observation.

Former figure 2f:

Monoisotopic amino acid weights:

I-letter code	3-letter code	Chemical formula	Monoisotopic	Average
A	Ala	C ₃ H ₅ ON	71.03711	71.0788
R	Arg	C ₆ H ₁₂ ON ₄	156.10111	156.1875
N	Asn	C ₄ H ₆ O ₂ N ₂	114.04293	114.1038
D	Asp	C ₄ H ₅ O ₃ N	115.02694	115.0886
C	Cys	C ₃ H ₅ ONS	103.00919	103.1388
E	Glu	C ₅ H ₇ O ₃ N	129.04259	129.1155
Q	Gln	C ₅ H ₈ O ₂ N ₂	128.05858	128.1307
G	Gly	C ₂ H ₃ ON	57.02146	57.0519
H	His	C ₆ H ₇ ON ₃	137.05891	137.1411
I	Ile	C ₆ H ₁₁ ON	113.08406	113.1594
L	Leu	C ₆ H ₁₁ ON	113.08406	113.1594
K	Lys	C ₆ H ₁₂ ON ₂	128.09496	128.1741
M	Met	C ₅ H ₉ ONS	131.04049	131.1926
F	Phe	C ₉ H ₉ ON	147.06841	147.1766
P	Pro	C ₅ H ₇ ON	97.05276	97.1167
S	Ser	C ₃ H ₅ O ₂ N	87.03203	87.0782
T	Thr	C ₄ H ₇ O ₂ N	101.04768	101.1051
W	Trp	C ₁₁ H ₁₀ ON ₂	186.07931	186.2132
Y	Tyr	C ₉ H ₉ O ₂ N	163.06333	163.1760
V	Val	C ₅ H ₉ ON	99.06841	99.1326

8. Immunopeptide enrichment in methods: DTT was probably not used for the crosslinking. I assume it should read DMP.

Thank you for your feedback. We corrected this in the revised manuscript.

9. Figure 2b: While the figure does provide the intended information, this reviewer believes that the data could be presented in a more condensed way that is also quicker to grasp. Please consider changing the graph type.

Thank you for your feedback. We have averaged the values for each condition, and chose a new visualization that will hopefully make our data more clear to the reader. The new plot in Figure 2 now also contains the absolute number of scans.

10. Figure 4a: Add a small axis label on the very top left to indicate frequency of detection or similar

Thank you for your feedback. **We modified Fig. 4 as suggested.**

11. Respectfully is used a couple of times in the manuscript when it should read respectively. Please correct. (Remarks on code availability)

Thank you for bringing this to our attention. **We corrected this across the manuscript.**

Reviewer #3 (Remarks to the Author)

In this manuscript, Shapiro et al. describe a novel immunopeptidome data acquisition strategy that incorporates neoantigens inferred from external sources to enhance acquisition sensitivity while maintaining control over the total number of immunopeptidome scans. The authors demonstrate: (a) that a wider isolation window, when combined with MSFragger-DDA+, improves the handling of co-fragmented MS2 scans; (b) enhanced sensitivity in identifying clinically relevant neoantigens compared to conventional DDA, DDA with an inclusion list, and DIA approaches; (c) increased confidence in identified peptides relative to existing methods. I have the following comments for the authors to consider:

Majors:

[1] While I certainly appreciate the bottleneck in neoantigen identification in clinical settings and the efforts to address this challenge, I want to ensure I fully understand the innovation presented here. From my reading of the manuscript, it seems that the primary added utility of NeoDiscMS is the incorporation of a 'real-time search' step following scouting MS2. However, this step is executed by instrument-dependent software rather than NeoDiscMS itself, which primarily generates the inclusion list and FASTA file before resuming after data acquisition.

If my understanding is correct (please correct me if I am wrong), would it be more accurate to describe the contribution as advocating for the integration of real-time search into immunopeptidome analysis, rather than presenting NeoDiscMS as a novel pipeline or extension?

Thank you for your feedback. NeoDiscMS describes an LC-MS immunopeptidomics method for personalized clinical T-cell antigen discovery workflows. It is indeed not a pipeline. Since

NeoDiscMS extends the functionality of NeoDisc ⁵ (list of prioritized targets have now an additional use: they enhance target detection when used with NeoDiscMS), we do consider it an extension of NeoDisc. It is NeoDisc that prioritizes targets and produces the inclusion and FASTA file that serve as inputs to the NeoDiscMS mass spectrometry acquisition, allowing to personalize the method for each (patient) sample.

The innovative aspects of NeoDiscMS are:

- Elevating the personalization of clinical immunopeptidomics workflow to the data acquisition level by incorporating NeoDisc-prioritized antigens into the method.
- Maximizing immunopeptide target sensitivity while minimizing loss of global depth by:
 - o splitting the method tree into two MS2 scan modules: discovery branch and targeted branch
 - o using scouting scans and subsequent RTS filter to selectively trigger high sensitivity scan system in the targeted branch. Additionally, no dynamic exclusion applies to provoke repeated fragmentation to further increase identification confidence.
 - o using wider isolation windows to benefit more from the deconvolution of chimeric spectra for more global depth ^{3,4}.
- Does not rely on spike-ins or prior runs to define targets ⁶⁻⁹.

None of these aspects have been previously reported for clinical immunopeptidomics or immunopeptidomics in general. NeoDiscMS combines these elements successfully to provide enhanced performance to our pipeline that needs to operate under specific limitations such as low sample input and short turnaround time. The reviewer is correct about the critical importance of real-time searches to improve the performance of NeoDiscMS.

We designed NeoDiscMS to be simple and affective making it easier for the community to adopt. To that end, NeoDiscMS integrates available tools and concepts (e.g. NeoDisc ⁵ – somatic mutation calling and immunogenicity prediction, DeepLC ¹⁰ – retention time prediction, RTS – real-time database search, MSFragger-DDA+ ³ - chimeric spectrum deconvolution) into a method that addresses previously unmet needs for clinical immunopeptidomics.

Inspired by the reviewer’s feedback we added a comment in the introduction adding emphasis on the modularity of NeoDiscMS as this was not spelled out there as such yet but only in the discussion. The other points of innovation (spike-in free and scalable discovery with increased sensitivity, personalization through preceding NGS data processing, sMS2-RTSf-hMS2 system for selective triggering of high-sensitivity scans, use of wider isolation windows to increase depth with chimeric spectral deconvolutions) were already addressed in the introduction and discussion.

[2] Building on my previous point, while the authors mention in the discussion that this approach can be applied to virtually all MS instruments, there is no demonstration of its performance on platforms from other vendors, such as Bruker TIMS or SCIEX. Would the reported improvement in sensitivity necessarily extrapolate to these systems? It’s perfectly fine if not, but I believe it would be valuable to clarify this, as it would help researchers using different platforms make informed decisions about selecting the most appropriate acquisition method.

Thank you for your feedback.

We assume that the reviewer refers to the statement “Importantly, NeoDiscMS is technically compatible with any MS architecture capable of generating MS/MS spectra.” towards the end of the discussion.

The architecture of mass spectrometers refers to the respective hardware (the ion source, mass analyzers, detectors, electronics, etc.) of an MS instrument. The MS architecture that is required to use NeoDiscMS is indeed very basic: the capability of generating MS/MS spectra and simple computations under communication with an external computer (as is also necessary for regular DDA methods to select peaks or maintain dynamic exclusion lists, for instance). This stands in contrast to other methods using real-time search that require an additional mass analyzer + detector to run, or the capability of generating MS3 spectra.

Currently, real-time search is available for the acquisition method editor software for Tribrid Eclipse and Tribrid Ascend instruments by Thermo. The reviewer is right—NeoDiscMS cannot be directly implemented on most instruments with vendor-provided method editors, due to software limitations. This, not because of constraints of the mass spectrometer architecture, but rather due to limitations of vendor-provided software solutions. We appreciate that the quoted statement could, hence, confuse readers, nonetheless. **We replaced the statement with just a description of which instruments let users implement NeoDiscMS at the time of writing.** We hope that this study will encourage broader vendor adoption of this approach and the integration of real-time search (RTS) capabilities.

“At the time of writing, the instruments equipped with vendor-supported acquisition method editors that enable implementation of NeoDiscMS are the Orbitrap Eclipse and Orbitrap Ascend (part of Thermo Fisher Scientific’s Tribrid Mass Spectrometer series). We envision a broader vendor adoption of this approach in the future and the integration of advanced RTS capabilities.”

[3] Another conceptual question I have is regarding the trade-off between immunopeptidome depth and neoantigen identification. While the authors argue that reducing global immunopeptidome depth by selecting the most promising sMS2 is beneficial, I wonder whether this is truly a major concern, especially given the compelling results in Fig. 5, where the added benefit of hMS2 in generating more fragment ions is clear. Since this ultimately leads to the identification of more neoantigens, which is the primary goal of the application, wouldn't it be worthwhile to prioritize a deeper immunopeptidome?

Thank you for your feedback. The reviewer is correct that identifying neoantigens and other targets is a key goal of NeoDiscMS. The clinical value of confidently identifying targets such as neoantigens is clear^{5,11}. Yet, in our view, this does not render global depth irrelevant. Most importantly, clinical tumor samples are obtained through highly invasive procedures from patients coping with severe burden of illness and treatment. These individuals, along with their families, willingly contribute precious tissue samples. Out of respect for their contribution, we view it as an ethical obligation to extract the greatest possible value from these specimens. Given the demonstrated benefits of comprehensive data depth (see below), it would be irresponsible to forgo this opportunity and risk losing critical information—both for immediate therapeutic design and for advancing future research. We believe it is crucial for both our team and the broader research community to develop and implement methods that optimize the insights derived from these invaluable clinical samples.

The importance of global depth in immunopeptidomics, that informs on the global immunopeptidome of the tumor sample, has several reasons:

- As we show in suppl. Fig. 2f, the discovery branch also provides sensitivity for targets. Global depth directly supports target discovery.
- The discovery branch might identify relevant peptides that were not prioritized, such as canonical and non-canonical TAAs. This can be important for studies where new classes of antigens/different protein databases are used for research.
- Immunopeptidomics data serves as a window into the dynamics of antigen processing¹²⁻¹⁴. The immunopeptidome of tumor samples is actively studied to better understand antigen processing in tumors, which is a vulnerable pathway that is frequently altered by cancer for immune evasion^{15,16}. Immunopeptidomics data from primary tumor samples is the most valuable in this respect, as it enables the investigation of clinically relevant, real-world scenarios.
- Global depth also provides further ways to retrospectively analyze clinical cohorts to extract knowledge, such as understanding the role of PTMs in adaptive immunity, for instance.
- Data of the global immunopeptidome directly contributes to improve immunogenicity predictions. This can be done by referring to entire databases, or to investigating the immunopeptidome of a specific sample for personalized immunotherapy design^{5,17-20}.
- On a purely bioinformatical level, one of the signature advantages of NeoDiscMS is that it can be searched in a “high mass resolution DDA”-like setting with search engines. This means that the discovery branch will provide a lot of unbiased MS2 spectra that are PSMed allowing for target-decoy FDR calculations. Having a high number of unbiased PSMs facilitates:
 - o better FDR control due to a large variety and absolute number of PSMed peptides without the bias of sampling for particular peptides, as by our prioritization. This facilitates more robust statistics²¹⁻²³.
 - o better use of rescoring algorithms, which depend on extracting features of PSM-candidate peptides to optimize score calculation for the separation of target and decoy populations. The DDA branch is guaranteed to provide spectra from abundant peptides which have a higher chance of showing high-quality MS2 spectra where features can be learned from. Rescoring brings particular benefit to immunopeptidomics data, making this point especially noteworthy²⁴.

We thank the reviewer for bringing up the question of global depth that made us realize that it was necessary to better introduce these aspects in the manuscript. **We have, therefore, dedicated an entire paragraph to highlight the value of the discovery branch in the discussion to address the importance of global depth in clinical immunopeptidomics.**

[4] I also want to challenge the purported benefit of using a wider window scan. Conceptually, I agree that it can help account for precursor ions deviating from their theoretical m/z. However, in practice, co-fragmentation will inevitably lead to more complex MS2 spectra, which is not ideal. While deconvolution methods and secondary peptide searches are designed to address this unavoidable challenge, intentionally introducing this issue seems somewhat counterintuitive. I believe this point warrants further clarification.

Thank you for your feedback.

One key metric to benchmark a processing method for identification in LC-MS-based proteomics (which includes immunopeptidomics) is the ability to provide FDR controlled identifications of high quality^{21,22,25}. As shown in fig. 2, there is added value to using wider isolation windows when aiming to deconvolute chimeric spectra as we identify more unique, high quality immunopeptides. The quality metrics that we use to assess these identifications are binding affinity predictions and peptide length distribution, both of which are the two key parameters to assess data quality in immunopeptidomics, and both of which are orthogonal to the scoring by the search engine (as long as the search engine is agnostic to HLA restriction). The quality of the set of peptides identified with our wide-window MSFragger-DDA+ strategy is equal to the quality of peptides that we identify when we use regular DDA isolation window sized. It is worth noting that the wide window isolation strategy impacts only the discovery branch, and in addition, users can implement NeoDiscMS without enabling the wide window isolation strategy, should they choose to do so.

We appreciate the concern that more complex MS2 spectra are problematic when deconvoluting chimeric spectra. We would kindly ask the reviewer to reconsider his/her concerns, pointing to the publications that describe algorithms for chimeric spectrum deconvolution^{3,4} and why they work so well. We hope we could convince the reviewer with our benchmarking metrics that there is added benefit in our approach and chimeric spectrum deconvolution, confirming the findings the authors of the indicated algorithms provide.

[5] Perhaps this is a naive question, but I would like to better understand why NeoDiscMS is expected to have greater sensitivity than DDA with an inclusion list. While I see that it reduces depth and increases the confidence of identified peptides, wouldn't using the full inclusion list, conceptually, be more effective in maximizing sensitivity?

Thank you for your feedback and questions regarding the performance of NeoDiscMS compared to iLDDA.

Typically, when using inclusion lists, the number of peptide targets is in the 10s or 100s, not reaching the NeoDiscMS benchmarked scale of 1'500. There are examples of workflows that focus on just targeting (like PRM), however, these workflows disregard global depth which is a requirement to us. In general, the performance of NeoDiscMS compared to iLDDA regarding target sensitivity is equal or slightly better. The reason for the possibility to perform better than iLDDA is the fact that the absolute number and features of scans on target precursors in NeoDiscMS is slightly different from an equivalent iLDDA measurement. Several factors play a role in this regard:

- The proportion of precursors captured by the inclusion list that are actually the target precursor is very small. Using a high-sensitivity scan on a precursor that is not the target but an isobaric molecule does not enhance target sensitivity and wastes time that could be better allocated to the discovery branch. The scouting scans, combined with real-time search filters, effectively manage this scenario.
- High-sensitivity scans without any selectivity take up more absolute time than first assessing the potential of a precursor with a scouting scan. The time that NeoDiscMS gains by being selective about high-sensitivity scans can be spent on further MS2 scans of eluants. Each of these scans bears a chance to identify a target as well.
- When the targeted branch is activated in NeoDiscMS up to two MS2 scans are triggered, depending on the real-time search filters. For iLDDA it is always only one scan. As each scan bears the chance to get PSMed to the target peptide, we have either the same or

better chances for this with NeoDiscMS compared to iLDDA. In cases where both iLDDA and NeoDiscMS only perform one scan with the targeted branch, the real-time search yielded a bad result. This indicates that the identification of that precursor as the target peptide is very unlikely to begin with.

We added a comment in the results pointing to the fact that more than 85% of inclusion list hits turn out to be isobaric non-targets, as by the results of RTSf. We also expanded the discussion that explains why NeoDiscMS doesn't only suffer less from loss in global depth than iLDDA, but also why we see equal or slightly greater target sensitivity.

“More than 85% of the inclusion list hits during a measurement do not pass RTSf, indicating that they are isobaric non-targets (Supplementary Fig. 3c).”

We also added a comment in the section “Scalable enhanced sensitivity for targets” that quantifies the improved trade-off between loss in global depth and improved target sensitivity.

“With respect to the number of uniquely identified peptides, we note that on average, for every 1% of loss in global depth compared to DDA, we gain 2% in target sensitivity with iLDDA and 5.6% with NeoDiscMS. This difference in sensitivity trade-off quantifies the minimization of loss in global depth with help of RTSf very clearly.”

And in the discussion: “The edge of NeoDiscMS over iLDDA lies in more efficient application of high-sensitivity scans. Long injection times on every analyte that is isobaric to a target within the scheduled RT window becomes increasingly troublesome as inclusion lists are scaled: the number of targets is critical for inclusion list burden. NeoDiscMS significantly reduces time spent sampling off-target ions. This leaves more time within an acquisition cycle to sample ions of other target masses or with the discovery branch. An additional advantage of NeoDiscMS is of its ability to sample the same precursor up to three times per cycle, compared to a maximum of two times in iLDDA. Each MS2 scan contributes a spectrum that can yield a high-quality PSM. This increased sampling frequency explains the higher global depth and the equal or slightly improved target sensitivity observed with NeoDiscMS relative to iLDDA.”

[6] Could the authors expand on the description of the DIA section in both the results and methods sections? I am unclear on how the spectral library was generated and what is meant by a spectrum-centric and hybrid DIA analysis. Additionally, I am unsure how NeoDiscMS can be integrated with DIA and why it would conceptually outperform DIA. If the key advantages proposed for NeoDiscMS are wider isolation windows and the inclusion of peptides of interest, wouldn't DIA already be an ideal choice, albeit with the trade-off of increased complexity and deeper coverage?

Thank you for your feedback. We hereby clarify our descriptions and elaborate on the advantages of NeoDiscMS over DIA for clinical target discovery.

In a peptide-centric DIA analysis, a library of immunopeptides capturing features like retention times and fragmentation patterns (usually a sample-specific library generated from DDA data) is used to match DIA data²⁶.

In a spectrum-centric DIA analysis, the DIA MS2 spectra is interpreted directly, without relying on MS2 spectra library. The complex DIA spectra are deconvoluted and searched against a peptide database similar to DDA²⁶.

The principles of spectrum- and peptide-centric data processing methods are well established in the literature²⁶⁻²⁹.

The “hybrid” term in “hybrid library (search)” is a reference to the fact that multiple types of data (in our case NeoDiscMS and spectrum-centric search of DIA, for other projects it is often DDA and DIA) are used together to generate a hybrid library, which is then used to search DIA data in a peptide-centric fashion^{30,31}.

We define this use of ‘hybrid library’ and ‘hybrid library search’ when the term is used for the first time in the section “Direct clinical application” in the results.

We would like to clarify that NeoDiscMS does not generally outperform DIA. These two acquisition methods are substantially different and we therefore propose to combine the two, just like DDA and DIA are often combined to provide a library for deep peptide-centric DIA searches. We demonstrated the performance of NeoDiscMS and DIA individually, and in combination:

We measured the same immunopeptidomics sample with DIA and NeoDiscMS. The DIA file that was analyzed by itself in a spectrum-centric fashion yielded a relatively low number of peptide identifications, compared to the same sample measured with NeoDiscMS. This is visualized in Fig. 6a, where NeoDiscMS yielded around double uniquely identified peptides by FragPipe compared to DIA with a spectrum-centric search (DIA-NN). As such, NeoDiscMS outperforms DIA when processed with a spectrum-centric search in terms of depth, and also in terms of target detection sensitivity, as can be seen in Fig. 6c. **We added a comment to the first paragraph of the sub-chapter “Direct clinical application” in the results that highlight this point.**

DIA is an attractive method achieving great depth and sensitivity when provided with a deep library to perform a peptide-centric search³². NeoDiscMS is therefore capable of generating sample-specific deep library that contains a high number high-confidence MS2 of our potential targets. **We added a comment to section “Direct clinical application” to highlight this point.**

An alternative to experimentally-derived libraries are computationally predicted libraries. This approach holds great promise, yet its performance in immunopeptidomics compared to empirical libraries is situational³²⁻³⁴ and accurate FDR calculations might face processing-specific challenges that need to be evaluated, just as is the case for spectrum-centric searches^{23,25}. **We added a comment to the discussion to highlight this point.**

Another important reason to not solely rely on DIA-derived peptide identifications (regardless of spectrum-centric or peptide-centric (with empirical or predicted libraries) processing) is the extreme complexity of DIA MS2 spectra. As already discussed above, when we want to identify neoantigens for patient interventions, we want to be as confident as possible about associating a spectrum with a peptide. For this we often resort to considering how spectra were processed and identified, and we also manually and visually inspect the spectra annotation. DIA MS2 spectrum annotation where precursors were isolated with a window of 20Th is much more ambiguous than an MS spectrum that was identified with an isolation window that is an order of magnitude smaller, because it will contain significantly less co-eluting precursor information that could lead

to random matching and skew fragment intensities. Having experimental evidence for a narrow-window MS2 is highly conducive to gain confidence in a PSM. If developments in the field of proteomics such as narrow-window DIA³⁵ might shift this balance remains to be seen. Therefore, we advocate for combining NeoDiscMS with DIA, to benefit as much from these complementary methods.

We added a comment to the discussion that highlights.

Minors:

[1] The term “Tissue ID” is inconsistently formatted throughout the text, sometimes as “Ti” and other times as “TI”, I suggest standardizing it for clarity and consistency.

Thank you for your feedback. We did indeed find one location where we spelled out “TI3” and not “Ti3”. **We changed this accordingly.**

[2] I suggest reconsidering the terms 'global depth' and 'global presentation patterns,' as their meaning is not immediately clear. It took me several reads to fully grasp their intended meaning. Perhaps a more precise alternative, such as 'total number of scans,' would improve clarity.

Thank you for your feedback.

We agree that these terms can be confusing. **We hence substituted the term “global presentation patterns” with “hotspot features”^{19,36} , a term that is known from the immunopeptidomics literature.**

We also added what we mean by global depth with the first use in the abstract (“the entirety of the measurable immunopeptidome”).

[3] The introduction, as well as parts of the results, seem to assume that readers have a deep background in immunopeptidomics and mass spectrometry. I suggest providing brief explanations of key concepts, such as resolution, dynamic exclusion, chimeric spectra deconvolution, to make the text more accessible to a broader audience.

Thank you for your feedback. We would like to address each of those terms point by point in the manuscript. Please find the according changes as indicated for each keyword.

- Resolution : we use this term in the methods section to describe mass spectrum acquisition parameters. We clarified what resolution is meant by assuring that we refer to “mass resolution”.
- Dynamic exclusion : We added a comment that explains the term where we mention “dynamic exclusion” for the first time in the introduction.
- Chimeric spectra deconvolution : We added a comment that explains the term “chimeric spectra” and “chimeric spectra deconvolution” where we mention “chimeric spectra deconvolution” for the first time in the introduction.

1. Yu, F. *et al.* Fast Quantitative Analysis of timsTOF PASEF Data with MSFragger and IonQuant. *Molecular & Cellular Proteomics* **19**, 1575–1585 (2020).
2. Kong, A. T., Leprevost, F. V., Avtonomov, D. M., Mellacheruvu, D. & Nesvizhskii, A. I. MSFragger: ultrafast and comprehensive peptide identification in mass spectrometry-based proteomics. *Nat Methods* **14**, 513–520 (2017).
3. Yu, F., Deng, Y. & Nesvizhskii, A. I. MSFragger-DDA+ enhances peptide identification sensitivity with full isolation window search. *Nat Commun* **16**, 3329 (2025).
4. Frejno, M. *et al.* Unifying the analysis of bottom-up proteomics data with CHIMERYS. *Nat Methods* (2025) doi:10.1038/s41592-025-02663-w.
5. Huber, F. *et al.* A comprehensive proteogenomic pipeline for neoantigen discovery to advance personalized cancer immunotherapy. *Nat Biotechnol* (2024) doi:10.1038/s41587-024-02420-y.
6. Stopfer, L. E. *et al.* Absolute quantification of tumor antigens using embedded MHC-I isotopologue calibrants. *Proc. Natl. Acad. Sci. U.S.A.* **118**, e2111173118 (2021).
7. Stopfer, L. E., Mesfin, J. M., Joughin, B. A., Lauffenburger, D. A. & White, F. M. Multiplexed relative and absolute quantitative immunopeptidomics reveals MHC I repertoire alterations induced by CDK4/6 inhibition. *Nat Commun* **11**, 2760 (2020).
8. Leddy, O. *et al.* Validation and quantification of peptide antigens presented on MHCs using SureQuant. *Nat Protoc* (2024) doi:10.1038/s41596-024-01076-x.
9. Pollock, S. B. *et al.* Sensitive and Quantitative Detection of MHC-I Displayed Neoepitopes Using a Semiautomated Workflow and TOMAHAQ Mass Spectrometry. *Molecular & Cellular Proteomics* **20**, 100108 (2021).
10. Bouwmeester, R., Gabriels, R., Hulstaert, N., Martens, L. & Degroeve, S. DeepLC can predict retention times for peptides that carry as-yet unseen modifications. *Nat Methods* **18**, 1363–1369 (2021).

11. Lang, F., Schrörs, B., Löwer, M., Türeci, Ö. & Sahin, U. Identification of neoantigens for individualized therapeutic cancer vaccines. *Nat Rev Drug Discov* **21**, 261–282 (2022).
12. Shapiro, I. E. *et al.* Deleterious KOs in the HLA Class I Antigen Processing and Presentation Machinery Induce Distinct Changes in the Immunopeptidome. *Molecular & Cellular Proteomics* **24**, 100951 (2025).
13. McShan, A. C. *et al.* TAPBPR promotes antigen loading on MHC-I molecules using a peptide trap. *Nat Commun* **12**, 3174 (2021).
14. Marijt, K. A. *et al.* Identification of non-mutated neoantigens presented by TAP-deficient tumors. *Journal of Experimental Medicine* **215**, 2325–2337 (2018).
15. Galassi, C., Chan, T. A., Vitale, I. & Galluzzi, L. The hallmarks of cancer immune evasion. *Cancer Cell* **42**, 1825–1863 (2024).
16. Dunn, G. P., Bruce, A. T., Ikeda, H., Old, L. J. & Schreiber, R. D. Cancer immunoediting: from immunosurveillance to tumor escape. *Nat Immunol* **3**, 991–998 (2002).
17. Pyke, R. M. *et al.* Precision Neoantigen Discovery Using Large-Scale Immunopeptidomes and Composite Modeling of MHC Peptide Presentation. *Molecular & Cellular Proteomics* **22**, 100506 (2023).
18. Tokita, S. *et al.* Identification of immunogenic HLA class I and II neoantigens using surrogate immunopeptidomes. *Sci. Adv.* **10**, eado6491 (2024).
19. Müller, M., Gfeller, D., Coukos, G. & Bassani-Sternberg, M. ‘Hotspots’ of Antigen Presentation Revealed by Human Leukocyte Antigen Ligandomics for Neoantigen Prioritization. *Front. Immunol.* **8**, 1367 (2017).
20. Kraemer, A. I. *et al.* The immunopeptidome landscape associated with T cell infiltration, inflammation and immune editing in lung cancer. *Nat Cancer* **4**, 608–628 (2023).
21. Elias, J. E. & Gygi, S. P. Target-Decoy Search Strategy for Mass Spectrometry-Based Proteomics. in *Proteome Bioinformatics* (eds. Hubbard, S. J. & Jones, A. R.) vol. 604 55–71 (Humana Press, Totowa, NJ, 2010).

22. Gupta, N., Bandeira, N., Keich, U. & Pevzner, P. A. Target-Decoy Approach and False Discovery Rate: When Things May Go Wrong. *J. Am. Soc. Mass Spectrom.* **22**, 1111–1120 (2011).
23. Wen, B. *et al.* Assessment of false discovery rate control in tandem mass spectrometry analysis using entrapment. Preprint at <https://doi.org/10.1101/2024.06.01.596967> (2024).
24. Yang, K. L. *et al.* MSBooster: improving peptide identification rates using deep learning-based features. *Nat Commun* **14**, 4539 (2023).
25. Nesvizhskii, A. I. Proteogenomics: concepts, applications and computational strategies. *Nat Methods* **11**, 1114–1125 (2014).
26. Ting, Y. S. *et al.* Peptide-Centric Proteome Analysis: An Alternative Strategy for the Analysis of Tandem Mass Spectrometry Data. *Molecular & Cellular Proteomics* **14**, 2301–2307 (2015).
27. Tsou, C.-C. *et al.* DIA-Umpire: comprehensive computational framework for data-independent acquisition proteomics. *Nat Methods* **12**, 258–264 (2015).
28. Li, K., Teo, G. C., Yang, K. L., Yu, F. & Nesvizhskii, A. I. diaTracer enables spectrum-centric analysis of diaPASEF proteomics data. *Nat Commun* **16**, 95 (2025).
29. Guo, T. & Aebersold, R. Recent advances of data-independent acquisition mass spectrometry-based proteomics. *Proteomics* **23**, 2200011 (2023).
30. Lapcik, P. *et al.* A hybrid DDA/DIA-PASEF based assay library for a deep proteotyping of triple-negative breast cancer. *Sci Data* **11**, 794 (2024).
31. Willems, P., Fels, U., Staes, A., Gevaert, K. & Van Damme, P. Use of Hybrid Data-Dependent and -Independent Acquisition Spectral Libraries Empowers Dual-Proteome Profiling. *J. Proteome Res.* **20**, 1165–1177 (2021).
32. Pak, H. *et al.* Sensitive Immunopeptidomics by Leveraging Available Large-Scale Multi-HLA Spectral Libraries, Data-Independent Acquisition, and MS/MS Prediction. *Molecular & Cellular Proteomics* **20**, 100080 (2021).

33. Wahle, M. *et al.* IMBAS-MS Discovers Organ-Specific HLA Peptide Patterns in Plasma. *Molecular & Cellular Proteomics* **23**, 100689 (2024).
34. Oliynyk, D. *et al.* diaPASEF analysis for HLA-I peptides enables quantification of common cancer neoantigens. *Molecular & Cellular Proteomics* 100938 (2025)
doi:10.1016/j.mcpro.2025.100938.
35. Guzman, U. H. *et al.* Ultra-fast label-free quantification and comprehensive proteome coverage with narrow-window data-independent acquisition. *Nat Biotechnol* **42**, 1855–1866 (2024).
36. Jappe, E. C., Kringelum, J., Trolle, T. & Nielsen, M. Predicted MHC peptide binding promiscuity explains MHC class I ‘hotspots’ of antigen presentation defined by mass spectrometry eluted ligand data. *Immunology* **154**, 407–417 (2018).